# Glia control experience-dependent plasticity in an olfactory critical period

Hans C Leier[1†], Alexander J Foden[1†], Darren A Jindal[1], Abigail J Wilkov[1], Paola Van der Linden Costello[1], Pamela J Vanderzalm[2], Jaeda Coutinho-Budd[3], Masashi Tabuchi[1], Heather T Broihier[1]*

[1]Department of Neurosciences, Case Western Reserve University School of Medicine, Cleveland, United States; [2]Department of Biology, John Carroll University, University Heights, United States; [3]Department of Neuroscience, University of Virginia School of Medicine, Charlottesville, United States

*For correspondence:
heather.broihier@case.edu

†These authors contributed equally to this work

## eLife Assessment

Periods in which experience regulates early plasticity in sensory circuits are well established, but the mechanisms that control these critical periods are poorly understood. In this **important** study, the authors examine early-life critical periods that regulate the *Drosophila* antennal lobe and show that constant odor exposure markedly reduces the volume, synapse number, and function of a specific glomerulus. The authors offer **compelling** evidence that these changes are mediated by the invasion of ensheathing glia into the glomerulus where they phagocytose connections via a mechanism involving the engulfment receptor Draper.

**Abstract** Sensory experience during developmental critical periods has lifelong consequences for circuit function and behavior, but the molecular and cellular mechanisms through which experience causes these changes are not well understood. The *Drosophila* antennal lobe houses synapses between olfactory sensory neurons (OSNs) and downstream projection neurons (PNs) in stereotyped glomeruli. Many glomeruli exhibit structural plasticity in response to early-life odor exposure, indicating a general sensitivity of the fly olfactory circuitry to early sensory experience. We recently found that glia shape antennal lobe development in young adults, leading us to ask if glia also drive experience-dependent plasticity during this period. Here, we define a critical period for structural and functional plasticity of OSN-PN synapses in the ethyl butyrate (EB)-sensitive glomerulus VM7. EB exposure for the first 2 days post-eclosion drives large-scale reductions in glomerular volume, presynapse number, and post- synaptic activity. Crucially, pruning during the critical period has long-term consequences for circuit function since both OSN-PN synapse number and spontaneous activity of PNs remain persistently decreased following early-life odor exposure. The highly conserved engulfment receptor Draper is required for this critical period plasticity as ensheathing glia upregulate Draper, invade the VM7 glomerulus, and phagocytose OSN presynaptic terminals in response to critical-period EB exposure. Loss of Draper fully suppresses the morphological and physiological consequences of critical period odor exposure, arguing that phagocytic glia engulf intact synaptic terminals. These data demonstrate experience-dependent pruning of synapses and argue that *Drosophila* olfactory circuitry is a powerful model for defining the function of glia in critical period plasticity.

## Introduction

Critical periods are defined developmental windows in which sensory experience has an outsize influence on circuit function. Not only are sensory circuits uniquely plastic during these temporal windows, but the ensuing changes to circuit function are typically permanent (*Hensch, 2005*; *Levelt and*

*Hübener, 2012*). Critical period dysregulation is increasingly linked to neurodevelopmental disorders (*Gomes et al., 2019*; *LeBlanc and Fagiolini, 2011*), making the mechanisms that establish both timing and execution of this plasticity of great interest. Since ocular dominance plasticity in the mammalian visual cortex was established by Hubel and Wiesel as the preeminent model for the study of critical periods (*Hubel and Wiesel, 1970*; *Wiesel and Hubel, 1963*), great strides have been made to define the mechanisms of this form of critical period plasticity (*Hensch, 2005*; *Reh et al., 2020*; *Starkey et al., 2023*). However, the extent to which the specific mechanisms underpinning ocular dominance plasticity are generalizable and conserved among distinct circuit types is uncertain. Discovery of new molecular and cellular mechanisms required for critical period plasticity would be accelerated by the advent of a robust form of critical period plasticity in an accessible genetic model system.

The *Drosophila melanogaster* olfactory system has emerged as an attractive candidate in which to understand the role of sensory experience in circuit development. With a total of 61 unique olfactory receptors (ORs; *Mier et al., 2022*) expressed in 2281 olfactory sensory neurons (OSNs) (*Schlegel et al., 2021*) that terminate in stereotyped glomeruli in the antennal lobe, it shares the basic organization of the mammalian olfactory epithelium and bulb at a scale that is orders of magnitude less complex (*Mori et al., 1999*). Pioneering work by Ferrús and colleagues (*Devaud et al., 2003a*; *Devaud et al., 2001*) revealed structural changes and behavioral habituation in response to concentrated odorant exposure in early adulthood, similar to that observed in mammals (*Inoue et al., 2021*). Together, the exquisitely stereotyped organization of the fly antennal lobe together with the detailed mapping of its circuitry and the high degree of conservation with mammalian systems make the *Drosophila* olfactory circuit a powerful model for investigating early-life plasticity.

Subsequent studies yielded insights into the mechanisms by which developing circuits respond to sensory input (*Chodankar et al., 2020*; *Das et al., 2011*; *Golovin et al., 2019*; *Gugel et al., 2023*; *Sachse et al., 2007*). An emerging theme is that critical period plasticity in this model likely serves to preserve combinatorial odor encoding. Since many common odorants bind to several ORs with varying avidity (*Münch and Galizia, 2016*), the ability to discriminate among different odors (*Hallem and Carlson, 2006*), encode position (*Kadakia et al., 2022*; *Taisz et al., 2023*), and assign valence (*Knaden et al., 2012*; *Semmelhack and Wang, 2009*) relies on specific, balanced combinations of activity levels across various sensory neuron populations (*Si et al., 2019*). This code would be jeopardized by exposure to OR-saturating concentrations of a single odorant (*Tadres et al., 2022*), leading to different forms of adaptation in olfactory lobe circuitry. Short-term adaptations include temporarily blocking OSN firing (*Tadres et al., 2022*), while long-term odorant exposure in young adults induces circuit remodeling to increase inhibition of overactive sensory inputs (*Chodankar et al., 2020*; *Das et al., 2011*; *Sudhakaran et al., 2012*). Specifically, DM2 and DM5, two glomeruli that are moderately sensitive to the odor ethyl butyrate (EB), increase in size as the result of early-life odor exposure. This increase is likely caused by the growth of inhibitory synapses between local interneurons (LNs) and second-order projection neurons (PNs), leading to long-term habituation.

Recently, a novel form of early-life remodeling was described in VM7, another EB-responsive glomerulus. OSNs expressing the odorant receptor Or42a project to VM7 and are much more strongly activated by EB than either DM2 or DM5, meaning that Or42a OSNs are activated at lower EB concentrations (*Münch and Galizia, 2016*). Following early-life EB exposure, VM7 circuitry appears to dampen output not by increasing inhibition, but rather by removing the Or42a OSN axon terminals themselves. The underlying mechanism was first proposed to be axon terminal retraction *Golovin et al., 2021*; *Golovin et al., 2019*, but more recently shown to require Draper-mediated signaling in glia (*Baumann et al., 2024*; *Nelson et al., 2024*). Draper is the *Drosophila* homolog of *C. elegans* CED-1 and mammalian Jedi-1 and MEGF10 (*Freeman et al., 2003*; *Suzuki and Nakayama, 2007*; *Wu et al., 2009*; *Zhou et al., 2001*). These findings build on recent studies elucidating key functions for glia in regulating critical period timing and plasticity (*Ackerman et al., 2021*; *Lee et al., 2022*; *Ribot et al., 2021*); however, many crucial questions remain unanswered. First, while this form of glial-dependent remodeling requires Draper (*Nelson et al., 2024*), it is unknown whether it requires Draper-mediated synapse pruning, since phagocytosis was not assessed. This is an important issue because while Draper is an engulfment receptor, it also serves key non-phagocytic signaling roles (*Hsu et al., 2021*; *Kay et al., 2012*; *McPhee et al., 2010*; *Wang et al., 2023*). Second, does this example of glial-dependent remodeling represent a true critical period, or rather, does it reflect broad early-life plasticity that we and others have described (*Jindal et al., 2023*; *Paolicelli et al., 2011*)? Third,

critical periods are widely understood to imply long-term effects on circuit function; yet it is unknown whether there are physiological consequences to early-life remodeling of VM7. This question is particularly salient since Or42a OSN terminal volume appears to recover quickly following EB exposure (*Chodankar et al., 2020*; *Golovin et al., 2019*), raising the possibility that it represents a transient, entirely reversible, response to exposure to concentrated odorant.

Here, we demonstrate experience-dependent pruning of Or42a OSN axon terminals during a strictly delimited critical period. Or42a OSN terminals are remodeled following concentrated odorant exposure from 0 to 2 days of adulthood (days post eclosion (DPE)), but not immediately before or after. EB exposure during this period leads to loss of terminal axon arbors and their presynaptic specializations. Importantly, while Or42a OSN axon arbor morphology recovers following a five-day recovery period, presynaptic number does not, demonstrating long-term consequences of critical period odor exposure. Moreover, spontaneous activity of the postsynaptic PN partners of Or42a OSNs is dramatically reduced both immediately following odor exposure and following the recovery period, indicating persistent effects on odor processing. We find that critical period remodeling requires Draper signaling in ensheathing glia, a glial population that normally enwraps olfactory glomeruli in the antennal lobe. In ensheathing glia-specific *draper* RNAi animals, ensheathing glia do not invade the VM7 glomerulus, and Or42a OSNs are not pruned. Strikingly, odor exposure does not alter VM7 PN spontaneous activity in *draper* knockdown animals, supporting our morphological analysis and arguing that Or42a OSN terminals are intact following odor exposure and only removed by Draper-mediated engulfment. Together, these data demonstrate an essential role for Draper in ensheathing glial infiltration and synaptic engulfment, as well as long-term consequences of experience-dependent synaptic pruning during an early-life critical period.

## Results

### Early-life exposure to elevated ethyl butyrate erodes Or42a OSN connectivity and function

The antennal lobe (AL) is the first processing center for olfactory information in the *Drosophila* brain (*Couto et al., 2005*; *Fishilevich and Vosshall, 2005*). In the AL, presynaptic terminals of primary olfactory sensory neurons, OSNs, synapse with second-order PNs within discrete glomeruli to transmit olfactory information to higher-order brain regions. In addition to OSNs and PNs, multiple populations of LNs (local interneurons) modulate OSN-PN transmission, as well as communication among glomeruli (*Figure 1A*). The AL is also home to populations of glia, including ensheathing glia (EG) and astrocytes (*Doherty et al., 2009*; *Freeman, 2015*; *Wu et al., 2017*).

While the response of AL glia to OSN axon injury has been thoroughly investigated (*Herrmann et al., 2022*; *Logan et al., 2012*; *Ziegenfuss et al., 2012*), glial roles in synapse formation, maintenance, and pruning in the AL are less clear. We previously investigated the function of the highly conserved glial engulfment receptor Draper in regulating axon terminal morphology and synapse number of several OSN populations across early and mid-adulthood to test for a role for Draper in shaping circuitry at baseline (*Jindal et al., 2023*). Intriguingly, the phenotypic and cell-type-specific consequences of losing glial Draper are not constant over time. Instead, loss of Draper leads to larger increases in synapse number in young flies than in older flies, suggesting that Draper-mediated synapse refinement is particularly important during early life. These findings piqued our interest in a form of early-life plasticity described in the VM7 glomerulus (*Chodankar et al., 2020*; *Golovin et al., 2021*; *Golovin et al., 2019*). In this paradigm, animals exposed to concentrated ethyl butyrate (EB) for the first two days of adulthood exhibit a near complete loss of Or42a OSN terminal arbors, suggesting the possibility of glial-mediated phagocytosis. This remodeling is suggested to reflect a new form of critical period plasticity; however, the EB concentrations typically used in these studies (25% [v/v]) also drive loss of terminal arbors when odor exposure occurs in mid-adulthood (*Golovin et al., 2019*), raising questions about critical period specificity. Moreover, the significance of this plasticity is uncertain, since Or42a terminals exhibit morphological recovery after animals are returned to a normal olfactory environment (*Chodankar et al., 2020*; *Golovin et al., 2019*).

We wanted to better describe the timing, scope, selectivity, and possible long-term consequences of this striking form of early-life plasticity before interrogating a role for glia. To begin, we employed our Imaris-based pipelines to quantify presynaptic number and OSN terminal arbor volume (*Jindal*

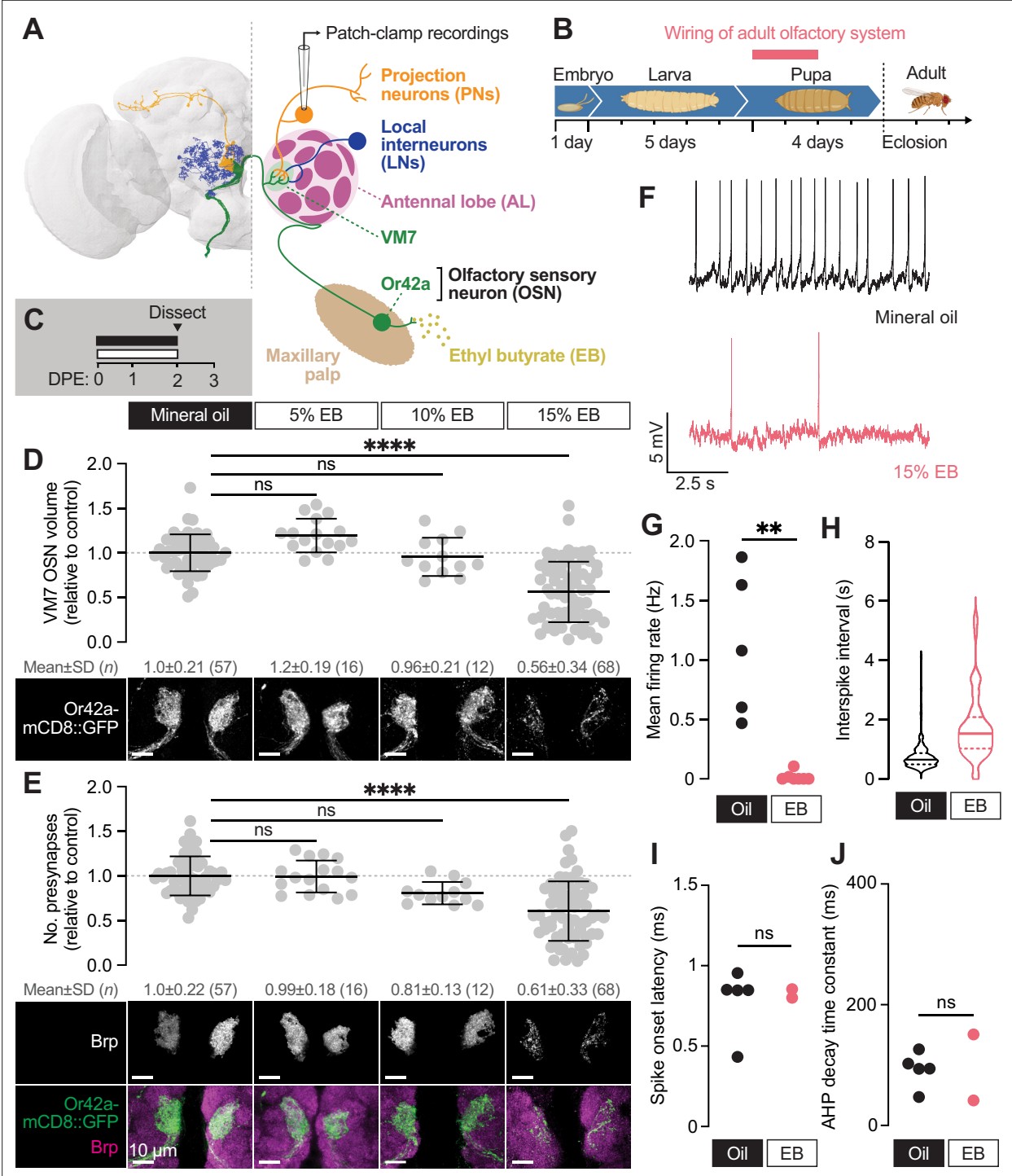

**Figure 1.** Early-life exposure to elevated ethyl butyrate erodes Or42a OSN connectivity and function. (**A**) FlyWire full-brain connectome reconstruction (left) and simplified schematic (right) of the Or42a olfactory circuit. Thirty-three olfactory sensory neurons (OSNs) expressing the ethyl butyrate (EB)-sensitive odorant receptor Or42a synapse with projection neurons (PNs) in a single glomerulus (VM7) of the antennal lobe (AL). VM7 also receives lateral input from local interneurons (LNs). (**B**) Overview of the *Drosophila* developmental timeline. OSNs synapse with PNs during the first 48 hr after puparium formation. (**C**) Schematic of the odorant exposure paradigm used in (**D–J**). White and black bars represent ethyl butyrate (EB) or mineral oil vehicle control, respectively. DPE, days post-eclosion. (**D**) Representative maximum intensity projections (MIPs) (bottom) and volume measurements (top) of the Or42a-mCD8::GFP OSN terminal arbor in VM7 in 2 DPE flies exposed to mineral oil or the indicated concentrations of EB throughout adulthood. (**E**) Representative MIPs (bottom) and number of VM7 presynapses (top) in 2 DPE flies exposed to mineral oil or the indicated concentrations of EB throughout adulthood. Presynapses were visualized with nc82 anti-bruchpilot (Brp) staining. Data are mean ± SD. (**F–J**) Patch-clamp recordings of VM7

*Figure 1 continued on next page*

*Figure 1 continued*

PNs from 2 DPE flies exposed to 15% EB or mineral oil throughout adulthood. (**F**) Representative PN membrane potential traces, (**G**) spontaneous mean firing rate (Oil, 1.1±0.61 [mean ± SD]; EB, 0.017±0.039), (**H**) interspike interval (ISI) events (Oil, 751.9±450.6; EB, 1757±1114), (**I**) spike onset latency (Oil, 0.79±0.21; EB, 0.83±0.038), and (**J**). afterhyperpolarization (AHP) decay time constant (Oil, 92.7±28.8; EB, 96.1±77.4). ns, not significant,. **p<0.01, ***p<0.001, ****p<0.0001, Kruskal-Wallis test with Dunn's multiple comparisons test (**D**) or Mann-Whitney U-test (**F–I**). Genotypes, raw values, and detailed statistics are provided in *Figure 1—source data 1*.

The online version of this article includes the following source data for figure 1:

**Source data 1.** Genotypes, raw values, and statistics for experiments shown in *Figure 1*.

*et al., 2023*) in young (0–2 DPE) animals across a range of EB concentrations. Briefly, we precisely masked Or42a-mCD8::GFP-labeled OSN terminal arbors using the Surfaces function in Imaris and quantified bruchpilot (Brp) puncta exclusively within the OSN mask. Brp is the fly homolog of mammalian ELKS/CAST/ERC and is a structural component of the presynaptic electron-dense T-bar (*Wagh et al., 2006*). We previously demonstrated that the number of presynapses is indistinguishable when quantified by masking endogenous Brp signal or by driving cell type-specific overexpression of Brp-short (*Jindal et al., 2023*), validating our approach. We focused on defining the lowest odorant concentration leading to robust reductions in VM7 OSN innervation volume and presynaptic number. We exposed 0–2 DPE animals to 5%, 10%, and 15% [v/v] EB or vehicle (mineral oil [MO]) and quantified presynapse number and Or42a terminal arbor volume (*Figure 1B–C*). Neither 5% or 10% EB significantly alters overall Or42a innervation volume or presynapse number (*Figure 1D–E*). In contrast, we observe a 44% loss of Or42a terminal arbors and a 39% reduction in presynapse number in animals exposed to 15% EB from 0 to 2 DPE (*Figure 1D–E*). Thus, 15% EB is sufficient to drive at least a short-term reduction in Or42a OSN innervation. The comparable decrease in OSN innervation volume and presynapse number suggests that there is not a selective removal of either presynaptic membrane or active zone components.

The loss of presynaptic release sites suggests fewer functional synapses, but whether innervation of PNs by Or42a OSNs is functionally reduced is unknown. To answer this question, we employed perforated patch-clamp electrophysiology to record spontaneous activity from PNs postsynaptic to Or42a OSNs (VM7 PNs; *Gouwens and Wilson, 2009*; *Kazama and Wilson, 2008*). Spontaneous activity of VM7 PNs was recorded ex vivo in brains in which the peripheral OSN axons were left intact, leading to persistence of spontaneous release (*Figure 1F*; Materials and methods; *Olsen and Wilson, 2008*).

We recorded spontaneous activity from VM7 PNs at 2–3 DPE following EB exposure (*Figure 1C*). Recordings were performed at least two hours after stopping odor exposure to ensure that our results were not confounded by short-term olfactory habituation. Strikingly, spontaneous activity is strongly decreased in VM7 PNs following 0–2 DPE EB exposure (*Figure 1F–H*). VM7 PNs were completely silent in 5 of the 7 EB-exposed brains we recorded from, while the frequency of firing in the two animals with residual synaptic activity was decreased 20-fold relative to MO-exposed brains (*Figure 1G–H*). To assess whether membrane properties of PNs are altered by EB exposure, we measured spike onset latency and the afterhyperpolarization (AHP) decay time constant. We find no difference in either metric in MO versus EB-exposed brains (*Figure 1I–J*), suggesting that intrinsic biophysical properties of these PNs are unchanged; additionally, previous studies have shown that OSN health (*Golovin et al., 2019*) and electrophysiology (*Devaud et al., 2001*) is generally unaffected by chronic odor exposure. Together, these data demonstrate that 0–2 DPE 15% EB exposure leads to a reduction of Or42a OSN innervation volume and presynaptic release sites (*Golovin et al., 2019*), as well as a striking near-complete loss of OSN-PN innervation.

## The first 2 days after eclosion is a critical period for Or42a OSNs

In *Drosophila*, late pupal development through early adult life represents an extended period of heightened structural and functional plasticity across the antennal lobe. Whereas OSN axon pathfinding and target recognition are largely completed after the first half of pupal development (*Komiyama and Luo, 2006*), the second half is characterized by a burst of synapse formation in many AL glomeruli (*Aimino et al., 2022*; *Aimino et al., 2023*; *Muthukumar et al., 2014*). Specifically, average Brp intensity in OSNs increases two- to threefold between 60–96hr after puparium formation (APF) and continues to increase through the first week of adulthood (*Aimino et al., 2022*). This prolonged period of synapse addition in olfactory circuits coincides with a period of heightened experience-dependent

synapse plasticity. In addition to VM7, other glomeruli (e.g. DM2, DM5, and V) exhibit structural plasticity following chronic odor exposure over the first week of adult life (*Chodankar et al., 2020*; *Das et al., 2011*; *Devaud et al., 2003b*; *Devaud et al., 2001*; *Sachse et al., 2007*).

Previous work suggests that OSN terminals in VM7 have become insensitive to chronic EB exposure by the second week of life (*Golovin et al., 2019*), but whether they can be remodeled throughout the first week is unknown. This is an important question given recent work demonstrating that synapse formation is widespread across the AL from mid-pupal stages through the first week of adulthood (*Aimino et al., 2022*), suggesting that this broad temporal domain represents a period of enhanced plasticity. Hence, we tested whether the structural plasticity observed by Or42a OSNs at 0–2 DPE reflects a more broadly defined sensitive period, or rather a well-defined critical period with a clear beginning and end. We exposed animals to 15% EB in 48hr intervals immediately before and after the 0–2 DPE window (*Figure 2A–I*). We found that exposing pupae to 15% EB for the last 2 days of pupal development (48hr-96hr APF) does not alter innervation volume or presynapse number of Or42a OSNs (*Figure 2A, D and G*). Thus, although this is a period of active synaptogenesis in the AL, synapse formation in late pupal Or42a OSNs does not appear to be influenced by environmental odorants, likely due in part to the physical barrier of the puparium limiting exposure to odorants and other sensory cues.

Next, we wondered for how many days after eclosion Or42a OSNs remain sensitive to EB exposure. We exposed animals to 15% EB from 2 to 4 DPE, a period during which many glomeruli (including VM7) are still in a phase of synapse addition (*Aimino et al., 2022*; *Jindal et al., 2023*) and assayed both terminal arbor size, via Or42a::mCD8-GFP, and presynapse number, as marked by Brp. We found that terminal arbor size and presynapse number in these animals is unchanged from those exposed to mineral oil over the same interval (*Figure 2C, F I*). Thus, the temporal window for experience-dependent plasticity of Or42a OSNs closes at 2 DPE and is unlikely to be a mere reflection of the fact that these terminals are normally in a phase of net synapse addition.

The temporal features of the critical period for Or42a OSNs are similar to those defined for other EB- sensitive glomeruli (*Chodankar et al., 2020*; *Das et al., 2011*; *Sachse et al., 2007*). Specifically, following 0–2 DPE EB exposure, the number of inhibitory synapses between PNs and LNs in EB-responsive glomeruli DM2 and DM5 increases. In other words, for these two glomeruli the increase in glomerular volume is proposed to result from more inhibitory LN1-PN synapses, without a change in the size of the OSN terminal itself (*Chodankar et al., 2020*). To further investigate the specificity of experience-dependent OSN pruning, we tested whether the OSNs innervating DM2 (Or22a OSNs), which also sense EB (*Figure 2J*), are pruned following early-life EB exposure. We find that Or22a OSN terminal arbor size is not altered upon EB exposure from 0 to 2 DPE (*Figure 2K*). This result is consistent with that observed for DM5 OSNs (*Chodankar et al., 2020*), and argues that the loss of primary sensory neuron innervation during the critical period is relatively specific to VM7. We hypothesize that the different forms of critical period plasticity observed in VM7 versus DM2 and DM5 may reflect distinct adaptations to chronic saturating odor exposure regulated by the different sensitivities of the respective OSNs to EB (see Discussion).

## The Or42a OSN-PN circuit does not recover from critical period pruning

Developmental critical periods are temporal windows when sensory experience drives long-term changes to the structure and function of synapses (*Hensch, 2005*). Thus, before calling the structural remodeling observed in VM7 from 0 to 2 DPE a true critical period, we wanted to establish if EB exposure causes persistent anatomical and/or functional effects. Or42a OSN terminal volume has been found to recover as soon as 12hr after EB exposure (*Chodankar et al., 2020*; *Golovin et al., 2019*), calling into question whether this striking anatomical remodeling is the product of a true critical period or a more transient form of time-limited plasticity. Thus, we set out to assess not only Or42a OSN terminal volume, but also presynapse number and PN activity following early-life EB exposure. In these experiments, we exposed 0–2 DPE flies to 15% EB as before, but instead of quantifying synaptic content or spontaneous PN activity immediately, we tested animals five days after the end of exposure (*Figure 3A*). In line with previous findings (*Chodankar et al., 2020*; *Golovin et al., 2019*), Or42a-positive OSN terminals return to their normal size after animals were returned to a normal rearing environment for five days (*Figure 3B*). In sharp contrast, the number of presynaptic

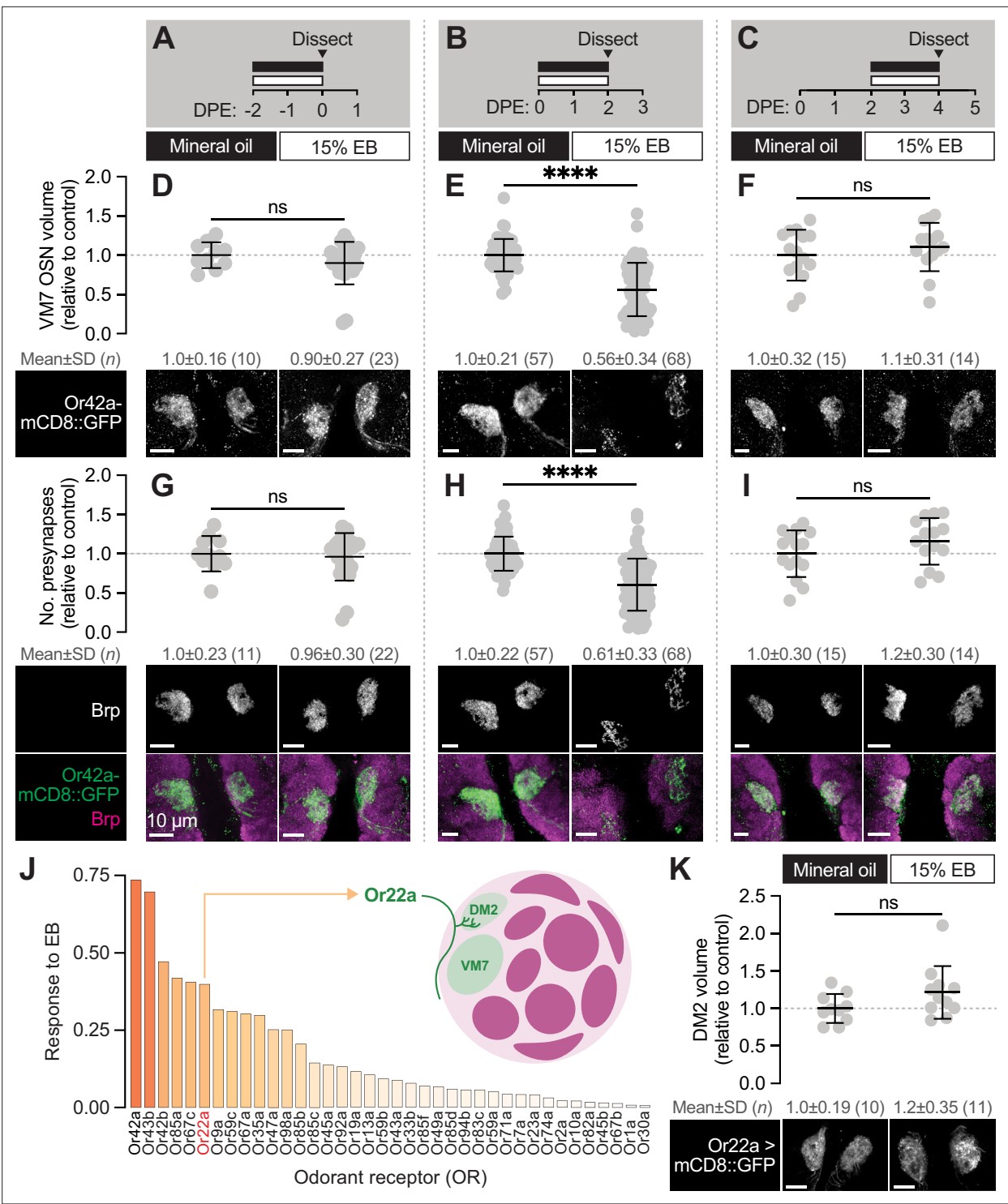

**Figure 2.** The first 2 days after eclosion is a critical period for Or42a OSNs. (**A–C**) Schematic of the odorant exposure paradigms used in (**D–I**). (**D–F**) Representative maximum intensity projections (MIPs; bottom) and volume measurements (top) of the Or42a-mCD8::GFP OSN terminal arbor in VM7 in flies exposed to mineral oil or 15% EB for the time periods indicated in (**A–C**). (**G–I**) Representative MIPs (bottom) and number of VM7 presynapses (top) in flies exposed to mineral oil or 15% EB for the time periods indicated in (**A–C**). Presynapses were visualized with nc82 anti-Brp staining. (**J**) Activation pattern of all glomeruli mapped in the DoOR 2.0 database to EB. Uncolored glomeruli are unmapped (dark gray) or nonresponsive to EB (light gray). (**K**) Representative MIPs (bottom) and volume measurements (top) of the Or22a OSN terminal arbor in DM2 in 2 DPE flies exposed to mineral oil or 15% EB throughout adulthood. Scale bar, 10 µm. Data are mean ± SD. ns, not significant, ****p <0.0001, Mann-Whitney U-test. Genotypes, raw values, and detailed statistics are provided in *Figure 2—source data 1*.

*Figure 2 continued on next page*

*Figure 2 continued*

The online version of this article includes the following source data for figure 2:

**Source data 1.** Genotypes, raw values, and statistics for experiments shown in *Figure 2*.

terminals within Or42a OSNs remains reduced by one-third (p=0.0056) in EB- vs. MO-exposed flies following the recovery period (*Figure 3C*); notably, these have been our only experiments to date where we observe a significant discordance between glomerular size and presynapse count. Thus, while OSN axon terminals repopulate the VM7 glomerulus during a recovery period following EB exposure, presynaptic terminals are not restored, opening the door to long-term functional effects of early-life EB exposure.

To test directly whether this unexpected result translates to long-term consequences to VM7 function, we employed perforated patch-clamp electrophysiology to record spontaneous activity from VM7 PNs (*Figure 3D*, schematic) following a five-day post-odor recovery period (*Figure 3A*). Strikingly, spontaneous activity of VM7 PNs remains persistently reduced after EB exposure (*Figure 3D–H*).

Specifically, the frequency of spontaneous activity in VM7 PNs is 8-fold lower than in mineral oil-exposed controls (*Figure 3E*), indicating that animals display long-lasting adaptations to concentrated odorant and dampen synaptic transmission downstream of Or42a OSNs. We do note that slightly fewer PNs are completely silenced than in animals immediately following odor exposure, perhaps as the result of LN plasticity during the recovery period. As in flies immediately following EB exposure, indices of PN membrane properties remain unaffected by EB exposure (*Figure 3G–H*), arguing that intrinsic biophysical properties of PNs are not altered by EB exposure. These electrophysiological findings are in line with the sustained reduction we find in presynaptic release site number (*Figure 3C*) and indicate that early-life odor exposure drives persistent structural and functional changes in the VM7 glomerulus. The long-lasting functional effects of odor exposure during the 0–2 DPE window indicate that this interval can be considered a critical period for development of olfactory circuitry.

## Glial Draper is required for Or42a experience-dependent pruning during its critical period

The loss of VM7 volume and synapses during critical period EB exposure suggests either retraction of Or42a axons or glial pruning of OSN presynaptic terminals. We recently demonstrated that glial Draper refines baseline morphology and synaptic content of several glomeruli, including VM7, during early adult life (*Jindal et al., 2023*). Draper's role in shaping olfactory circuitry positions it perfectly to facilitate remodeling in response to odor exposure during the critical period. Processes of two glial subtypes, ensheathing glia and astrocytes, ramify in the antennal lobe (*Figure 4A–B*). Ensheathing glia wrap individual glomeruli and are responsible for clearing axonal debris following OSN axotomy, while astrocyte processes are synapse-associated under homeostatic conditions and extensively infiltrate individual glomeruli (*Doherty et al., 2009*; *Freeman, 2015*). We previously found that Draper acts in both glial populations to shape glomerular volume and synapse number during normal development (*Jindal et al., 2023*).

To explore a function for Draper in experience-dependent synapse clearance, we began by knocking down *draper* expression in all glia via repo-GAL4. We used two validated *draper* RNAi lines in these experiments, one specific for the pro-phagocytic *draper-I* isoform, and the other predicted to target all three isoforms (*Logan et al., 2012*; *MacDonald et al., 2006*; *McPhee et al., 2010*). We exposed control animals to 15% EB from 0 to 2 DPE (*Figure 4C*) and found the characteristic loss of Or42a OSN terminal arbor volume following critical period odor exposure (*Figure 4D*). In contrast, pan-glial RNAi knockdown of either one or all *draper* isoforms completely suppresses the loss of Or42a OSN terminal arbor volume (*Figure 4D*). Thus, Draper acts in glia to remove terminal axonal arbors following critical period sensory experience. We next asked whether glial Draper also mediates the striking loss of Or42a OSN presynapses (*Figure 4E*). Indeed, pan-glial *draper* loss results in full restoration of Or42a OSN presynaptic content (*Figure 4E*).

These results indicate that loss of glial Draper suppresses the morphological remodeling observed in Or42a OSN terminals following critical period odor exposure. Does it also suppress the dramatic reduction in PN activity we observed following critical period odor exposure? To test the extent of Draper suppression, we carried out patch-clamp recordings from VM7 PNs in repo-GAL4 >UAS-*draper*

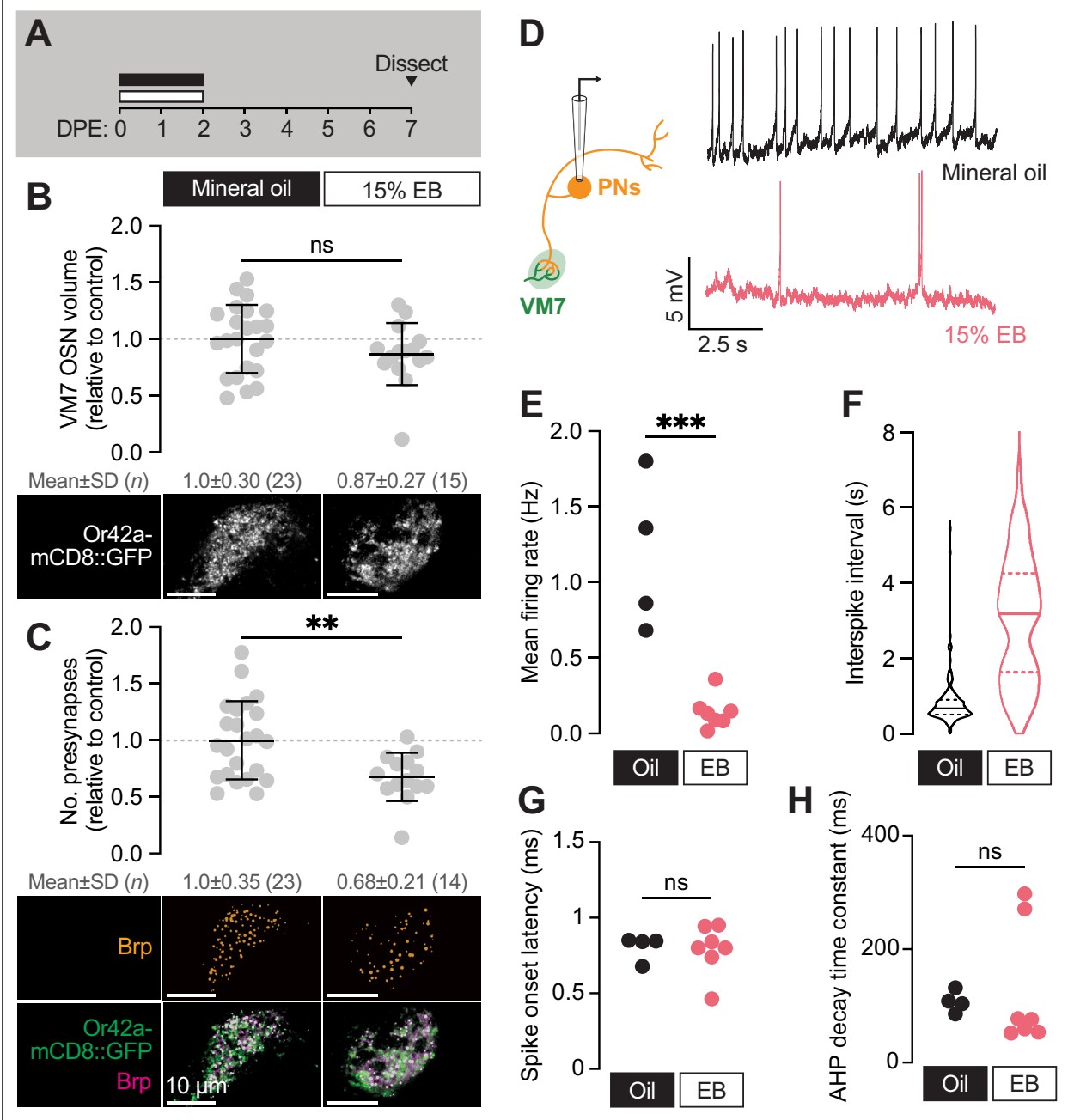

**Figure 3.** The VM7 OSN-PN circuit does not recover from critical period pruning. (**A**) Schematic of the odorant exposure paradigm used in (**B–H**). Flies were exposed to mineral oil or 15% EB during the critical period, then recovered without odorant until dissection at 7–8 DPE. (**B**) Representative MIPs (bottom) and volume measurements (top) of the Or42a-mCD8::GFP OSN terminal arbor in VM7. (**C**) Representative MIPs (bottom) and number of VM7 presynapses (top). Presynapses were visualized with nc82 anti-Brp staining. Data are mean ± SD. (**D–H**) Patch-clamp recordings of VM7 PNs from 7 to 8 DPE flies treated as in (**A**). (**D**) Schematic of patch-clamp recordings (left) and representative PN membrane potential traces (right), (**E**) spontaneous mean firing rate (Oil, 1.2±0.51; EB, 0.14±0.11), (**F**) interspike interval (ISI) events (Oil, 812.7±633.4; EB, 3027±1671), (**G**) spike onset latency (Oil, 0.80±0.084; EB, 0.79±0.16), and (**H**) afterhyperpolarization (AHP) decay time constant (Oil, 107.5±18.9; EB, 126.8±108.2). ns, not significant, **p<0.01, ***p<0.001, Mann-Whitney U-test. Genotypes, raw values, and detailed statistics are provided in *Figure 3—source data 1*.

The online version of this article includes the following source data for figure 3:

**Source data 1.** Genotypes, raw values, and statistics for experiments shown in *Figure 3*.

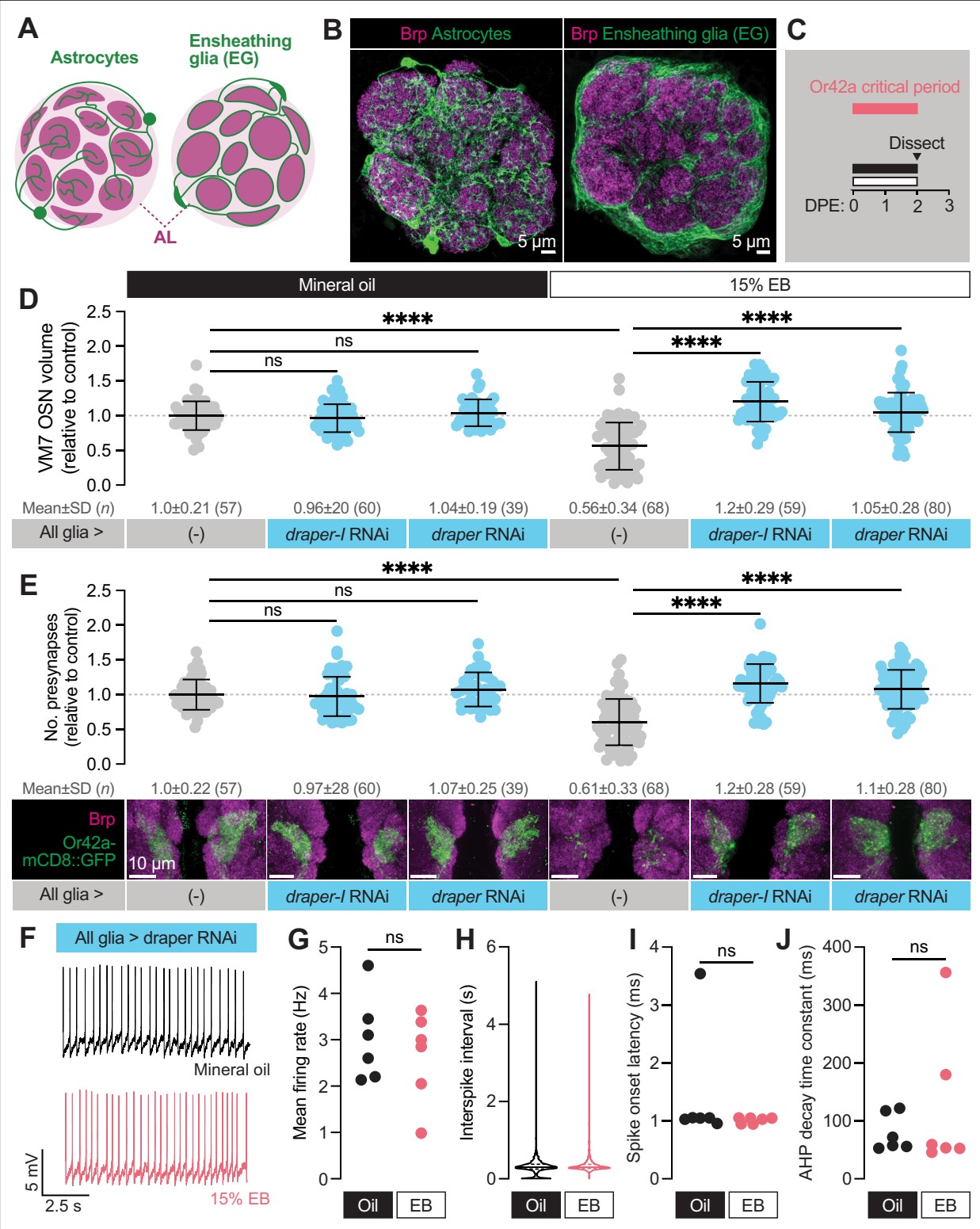

**Figure 4.** Glial Draper is required for Or42a activity-dependent pruning during its critical period. (**A–B**) Schematic (**A**) and representative MIPs (**B**) of astrocytes and ensheathing glia (EG), the two glial populations present in the neuropil of the antennal lobe. (**C**) Schematic of the odorant exposure paradigm used in **C, D** and *Figure 5*. (**D**) Representative MIPs (bottom) and volume measurements (top) of the Or42a-mCD8::GFP OSN terminal arbor in VM7. (**E**) Representative MIPs (bottom) and number of VM7 presynapses (top). Presynapses were visualized with nc82 anti-Brp staining. (**F–J**) Patch-clamp recordings of VM7 PNs from 2 DPE flies exposed to 15% EB or mineral oil throughout adulthood. (**F**) Representative PN membrane potential traces, (**G**) spontaneous mean firing rate (Oil, 3.0±0.93 [mean ± SD]; EB, 2.7±0.98), (**H**) interspike interval (ISI) events (Oil, 313.0±230.6; EB, 373.6±294.5),

*Figure 4 continued on next page*

*Figure 4 continued*

(**I**) spike onset latency (Oil, 1.4±1.0; EB, 1.0±0.05), and (**J**) afterhyperpolarization (AHP) decay time constant (Oil, 79.5±31.8; EB, 124.5±124.4). Data are mean ± SD. ns, not significant, ***p<0.001, ****p<0.0001, Kruskal-Wallis test with Dunn's multiple comparisons test. Pan-glial driver is repo-GAL4. Genotypes, raw values, and detailed statistics are provided in *Figure 4—source data 1*.

The online version of this article includes the following source data for figure 4:

**Source data 1.** Genotypes, raw values, and statistics for experiments shown in *Figure 4*.

RNAi animals (*Figure 4F–J*). Remarkably, in these animals there is no difference in spontaneous VM7 PN activity between flies exposed to MO or 15% EB from 0 to 2 DPE (*Figure 4G–H*), indicating that concentrated odor exposure during the critical period does not itself lead to deterioration of OSN terminals. Instead, presynaptic Or42a OSN terminals remain intact and functional following critical period odor exposure before being engulfed by glial Draper-mediated signaling.

Taken together, the suppression of Or42a OSN synapse loss by glial *draper* RNAi indicates that Draper-mediated signaling normally eliminates presynaptic terminals in response to critical period odor exposure. The lack of any observable change in OSN volume, presynaptic content, or physiology in *draper* knockdown animals argues that Draper is essential for critical period plasticity. Moreover, the finding that selective knockdown of the pro-phagocytic *draper* isoform is sufficient to block synapse and terminal arbor loss suggests a phagocytic mechanism. We investigate this hypothesis further below.

## Draper is required in ensheathing glia for the elimination of VM7 OSN presynaptic terminals

We previously uncovered a requirement for Draper in astrocytes and ensheathing glia in development of antennal lobe connectivity suggesting that both glial subtypes have phagocytic capability in this brain region. Interestingly, we found that Draper expression follows distinct time courses in the glial subtypes. In astrocytes, Draper expression peaks during late pupal development before decreasing in adulthood, while Draper is expressed at low levels in ensheathing glia during pupal development before increasing in adulthood (*Jindal et al., 2023*). The finding that Draper is predominantly expressed in ensheathing glia in the adult antennal lobe suggested that ensheathing glial Draper is responsible for clearing Or42a OSN terminals following critical period EB exposure. To test this hypothesis, we employed RNAi-mediated knockdown of *draper* specifically in astrocytes or ensheathing glia.

To reduce Draper expression specifically in astrocytes, we used alrm-GAL4 to drive *draper* RNAi (*Doherty et al., 2009*). In control animals (alrm-GAL4 >UAS-luciferase) exposed to 15% EB from 0 to 2 DPE, we found a 73% reduction in Or42a OSN presynaptic terminal volume (*Figure 5A*), in line with other control genotypes used in this study. Using the same *draper* RNAi constructs as before, we reduced *draper* expression specifically in astrocytes. In flies raised in normal conditions, we do not find a difference in Or42a terminal arbor volume in animals with astrocyte *draper* knockdown relative to controls at this time point (*Figure 5A*). Unlike pan-glial knockdown of *draper* expression (*Figure 4C*), knockdown of *draper* in astrocytes did not suppress Or42a OSN terminal volume loss (*Figure 5A*).

These results indicate that Draper is not required in astrocytes for activity-dependent remodeling of terminal axon arbors. Given that the mammalian Draper homolog MEGF10 acts in astrocytes to prune synapses (*Chung et al., 2013*), we also assessed whether astrocytic Draper might play a selective role in removing the synaptic structural protein Brp. However, we also failed to detect a difference in Brp puncta number in astrocyte *draper* knockdown animals relative to controls (*Figure 5B*). Together, these data suggest that Draper is not required in astrocytes for experience-dependent remodeling of olfactory circuitry, though astrocytes could certainly play essential Draper-independent roles in this process (see Discussion).

We next employed GMR56F03-GAL4 to knock down *draper* specifically in ensheathing glia (*Kremer et al., 2017*). As observed previously (*Jindal et al., 2023*), we did not find a role for ensheathing glial Draper in regulating basal Or42a OSN infiltration volume or presynaptic number (*Figure 5C*). In contrast, we see a clear requirement for ensheathing glial Draper in removing Or42a OSN axon terminals and synapses following critical period EB exposure (*Figure 5C–D*). Control animals bearing GMR56F03-GAL4 and UAS-luciferase transgenes exposed to 15% EB from 0 to 2 DPE displayed a

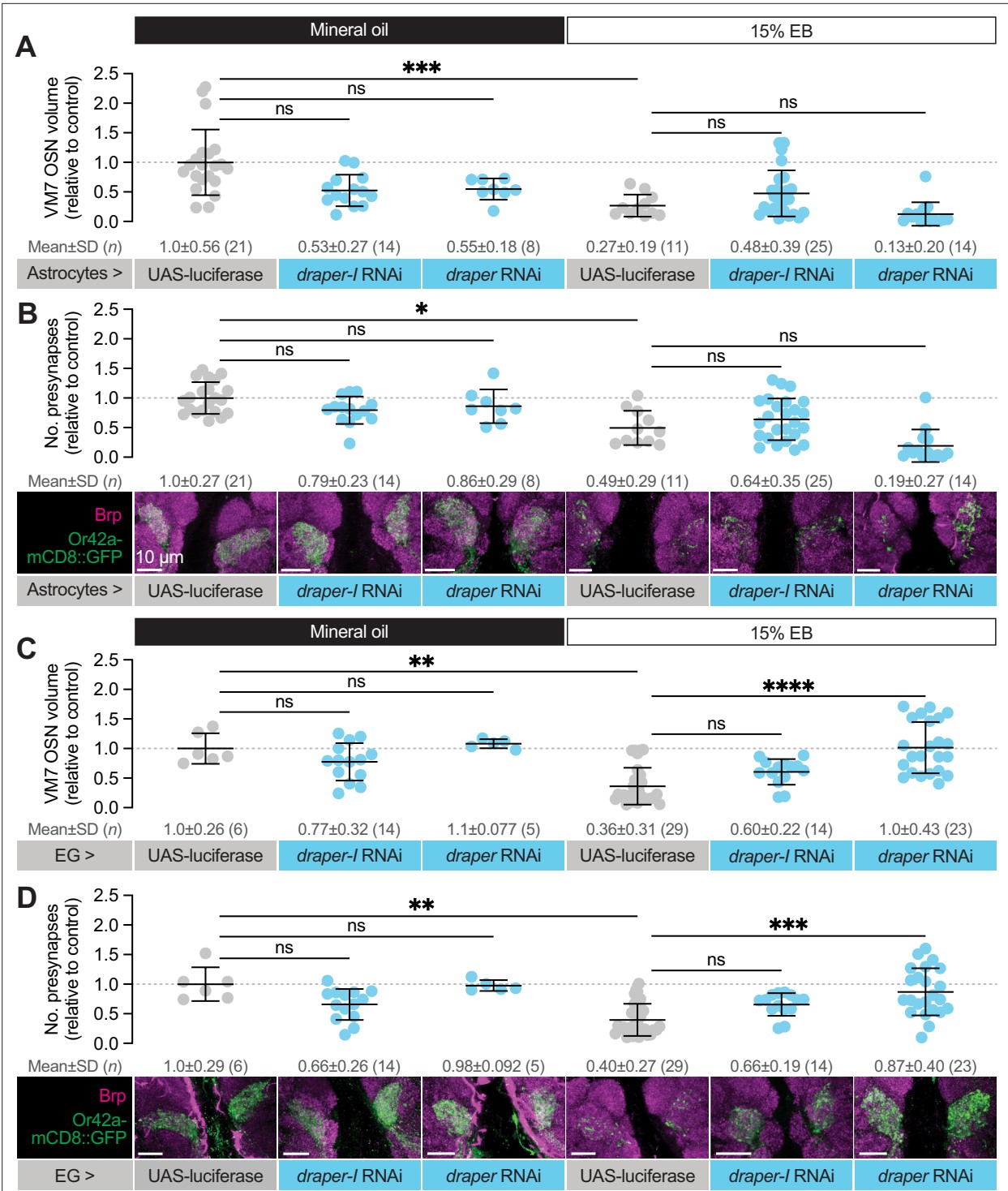

**Figure 5.** Draper is required in ensheathing glia to eliminate VM7 OSN presynaptic terminals. (**A**) Representative MIPs (bottom) and volume measurements (top) of the Or42a-mCD8::GFP OSN terminal arbor in VM7 in 2 DPE flies exposed to mineral oil or 15% EB throughout adulthood. (**B**) Representative MIPs (bottom) and number of VM7 presynapses (top). (**C**) Representative MIPs (bottom) and volume measurements (top) of the Or42a-mCD8::GFP OSN terminal arbor in VM7 in 2 DPE flies exposed to mineral oil or 15% EB throughout adulthood. (**D**) Representative MIPs (bottom) and number of VM7 presynapses (top). Presynapses were visualized with nc82 anti-Brp staining. Data are mean ± SD. ns, not significant, *p<0.05, **p<0.01, ***p<0.001, ****p<0.0001, Kruskal-Wallis test with Dunn's multiple comparisons test. Astrocyte driver is alrm-GAL4, EG driver is GMR56F03-GAL4. Genotypes, raw values, and detailed statistics are provided in *Figure 5—source data 1*.

The online version of this article includes the following source data and figure supplement(s) for figure 5:

*Figure 5 continued on next page*

*Figure 5 continued*

**Source data 1.** Genotypes, raw values, and statistics for experiments shown in *Figure 5*.

**Figure supplement 1.** No selective phagocytosis of presynapses during the Or42a critical period.

64% reduction in VM7 volume (*Figure 5C*) and a 60% reduction in Brp-positive presynaptic puncta (*Figure 5D*). RNAi-mediated *draper* knockdown in ensheathing glia suppresses this plasticity (*Figure 5C–D*). In GMR56F03>*draper* RNAi animals exposed to EB, axon terminals and presynaptic Brp puncta resemble that seen in animals exposed to mineral oil (*Figure 5C–D*). The full suppression of remodeling observed with ensheathing glial *draper* knockdown also indicates that astrocytic Draper cannot compensate for the loss of ensheathing glial Draper. We note that when the *draper* RNAi construct targeting a single isoform is used, we observe partial, non-significant, suppression of Or42a OSN remodeling. However, *draper-I* knockdown leads to a less punctate, more continuous glomerular appearance than seen in controls, and suppresses ensheathing glial infiltration of VM7 as well as the pan-isoform RNAi (*Figure 6*). Additionally, the negligible differences in presynaptic active zone density we observed across conditions and genotypes (*Figure 5—figure supplement 1*) support a model in which ensheathing glia phagocytose the OSN terminal arbor regardless of synaptic content. Together, these results argue that Draper-mediated signaling in ensheathing glia is responsible for removing axon terminals of VM7 OSNs following critical period odor exposure.

## Ensheathing glia extend processes into VM7 in a Draper-dependent manner

Draper is a highly conserved engulfment receptor, and its genetic requirement for synapse removal during critical period plasticity suggested the straightforward hypothesis (*Nelson et al., 2024*) that it mediates phagocytosis of Or42a OSN terminals. Before testing this, we investigated if ensheathing glial processes infiltrate VM7 following critical period EB exposure, and if so, whether this is Draper-dependent. This question is important because under normal circumstances, ensheathing glia wrap their processes around individual antennal lobe glomeruli (*Doherty et al., 2009*; *Freeman, 2015*), but do not extensively infiltrate them. However, they dramatically change their morphology following

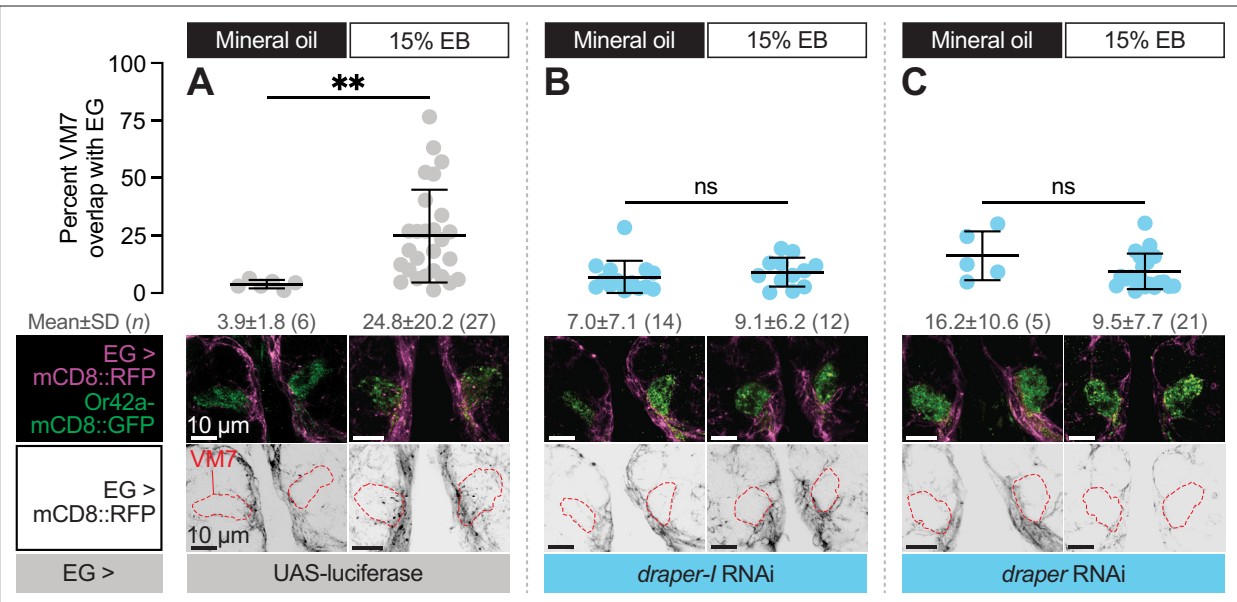

**Figure 6.** Ensheathing glia extend processes into VM7 to perform critical period pruning in a Draper-dependent manner. (**A–C**) Representative MIPs (bottom) and quantification (top) of percentage of VM7 volume (outline shown by dashed lines) occupied by ensheathing glial processes in 2 DPE flies exposed to 15% EB or mineral oil throughout adulthood. Data are mean ± SD. ns, not significant, **p<0.01, Mann-Whitney U-test. EG driver is GMR56F03-GAL4. Genotypes, raw values, and detailed statistics are provided in *Figure 6—source data 1*.

The online version of this article includes the following source data for figure 6:

**Source data 1.** Genotypes, raw values, and statistics for experiments shown in *Figure 6*.

injury to OSN axons. Severing OSN axons by surgically removing OSN cell bodies in the antennae or maxillary palps leads to Wallerian degeneration of antennal lobe distal axons and synapses (*Doherty et al., 2009*; *Freeman, 2015*; *MacDonald et al., 2006*). Following axotomy, ensheathing glia rapidly infiltrate affected glomeruli to engulf debris (*Doherty et al., 2009*). Given that ensheathing glia have the capacity to radically change their morphology in an injury context, we wondered whether they might also do so during the critical period.

To address this question, we simultaneously labeled ensheathing glial membranes with RFP (GMR56F03-GAL>UAS-mCD8::RFP) and knocked down *draper* expression in ensheathing glia (GMR56F03-GAL4>UAS-*draper* RNAi, UAS-*draper-I* RNAi, or UAS-luciferase [control]) in animals where VM7 OSN terminals are labeled with Or42a-mCD8::GFP. We exposed these animals to either mineral oil or 15% EB from 0 to 2 DPE and estimated infiltration by quantifying the percentage of total VM7 volume overlapping with the volume of ensheathing glial processes in Imaris. In mineral oil-exposed controls, we find that only 3.9% of VM7 volume is infiltrated by ensheathing glial processes (*Figure 6A*), consistent with the ensheathing function of this glial subtype. In stark contrast, 24.8% of VM7 volume overlaps with ensheathing glial membranes following critical period EB exposure (*Figure 6A*), suggesting that ensheathing glia have invaded VM7 and are thus well-poised to be directly involved in phagocytosing Or42a OSN axons and synapses. Following axotomy, loss of Draper abrogates ensheathing glial infiltration (*Doherty et al., 2009*) demonstrating that Draper-mediated signaling drives the morphological transformation of glia as well as elimination of debris. This dual requirement for Draper in the AL injury response prompted us to test whether it similarly drives ensheathing glial invasion during critical period remodeling. Indeed, *draper* knockdown in ensheathing glia prevents the infiltration observed following EB exposure in control animals (*Figure 6B–C*). Thus, Draper displays a temporally restricted requirement for ensheathing glial invasion into VM7 and removal of Or42a presynaptic terminals, in a manner that is highly similar to its requirement in the response to axon injury.

## Ensheathing glia upregulate Draper and phagocytose the terminal arbor of Or42a OSNs following critical period sensory experience

Draper is normally expressed on ensheathing glial membranes in the antennal lobe throughout early adult life (*Jindal et al., 2023*). The infiltration of ensheathing glial membranes into VM7 during critical period circuit remodeling suggests that Draper levels in the glomerulus were likely to increase in this condition. To test this hypothesis, we made use of a MiMIC line (Draper::GFP) in which the endogenous Draper locus is tagged with an EGFP-FlAsH-STREPII-TEV-3xFlag tag to permit tracking of endogenous protein expression (*Jindal et al., 2023*; *Nagarkar-Jaiswal et al., 2015*). We exposed Draper::GFP flies to mineral oil or EB during the critical period and quantified GFP intensity in VM7. In line with ensheathing glia infiltration into VM7 in response to EB exposure, we find a 66.7% increase in the mean intensity of Draper::GFP within VM7 in EB-exposed vs. mineral oil-exposed flies (*Figure 7A–B*). To investigate if Draper::GFP resides on infiltrating ensheathing glial membranes, we labeled them with RFP (GMR56F03-GAL4>UAS-mCD8::RFP). Indeed, we find clear colocalization of Draper::GFP with RFP (*Figure 7C*), showing that Draper is enriched on invading ensheathing glial membranes.

Taken together, these findings suggest that Draper is responsible for phagocytosis of Or42a OSN axon terminals by ensheathing glia following critical period odor exposure. To test this idea, we employed a sensor that takes advantage of the acidification in phagocytic compartments. We used MApHS (membrane-associated pH sensor), in which ecliptic pHluorin is fused to the extracellular side of CD4-tdTomato (*Han et al., 2014*; *Ji et al., 2023*). Ecliptic pHluorin is brightest at pH 7.5 but grows increasingly dim as the pH decreases. Because the fluorescence of tdTomato is not pH sensitive, the ratio of pHluorin/tdTomato has been used as a measure of Draper-mediated phagocytosis of the neurons expressing the sensor (*Ji et al., 2023*). We expressed MApHS in Or42a OSNs via Or42a-GAL4 and measured the pHluorin/tdTomato ratio in flies exposed to either mineral oil or EB during the critical period. Strikingly, we observe a loss of the pHluorin signal only in flies exposed to EB from 0 to 2 DPE, leading to a decrease in the pHluorin/tdTomato ratio (*Figure 7D*). This result is consistent with critical period elimination of Or42a OSN presynaptic terminals by phagocytosis. To test whether phagocytosis of Or42a OSN terminals requires Draper-mediated signaling in ensheathing glia, we again silenced *draper* via two independent RNAi lines. To permit independent control of

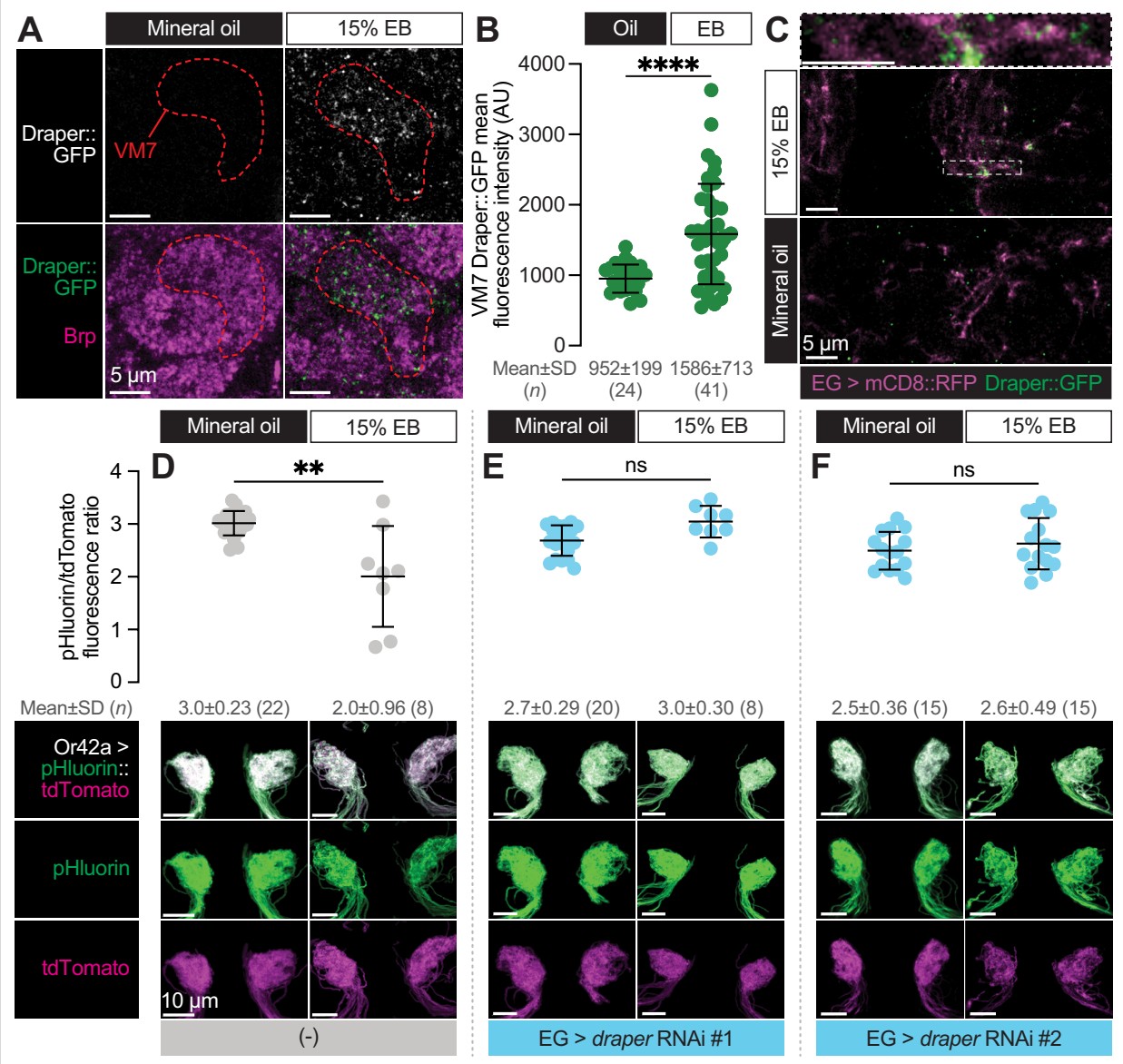

**Figure 7.** Ensheathing glia upregulate Draper in response to critical period Or42a activity and phagocytose its terminal arbor. (**A–C**) Representative single confocal planes of Brp staining with Draper (**A**), quantification of GFP mean fluorescence intensity within VM7 (**B**), and representative single confocal planes of EG membrane with Draper (**C**) of 2 DPE *draper::GFP* flies exposed to 15% EB or mineral oil throughout adulthood. Dashed lines indicate the outline of VM7. (**C**) (**D–F**) Representative MIPs (bottom) and quantification of pHluorin/tdTomato fluorescence intensity ratio (top) of 2 DPE flies exposed to 15% EB or mineral oil throughout adulthood. Data are mean ± SD. ns, not significant, \*\*p<0.01, \*\*\*\*p<0.0001, Mann-Whitney U-test. EG driver is *GMR56F03-LexA*. Genotypes, raw values, and detailed statistics are provided in *Figure 7—source data 1*.

The online version of this article includes the following source data for figure 7:

**Source data 1.** Genotypes, raw values, and statistics for experiments shown in *Figure 7*.

gene expression in sensory neurons and ensheathing glia, we employed two distinct binary expression systems. The GAL4-UAS system was again used to drive MApHS in Or42a OSNs (Or42a-GAL4>UAS-pHluorin::tdTomato) and the LexA-LexAOp system was used to silence *draper* in ensheathing glia (GMR56F03-LexA>LexAOp-*draper* RNAi; *Coutinho-Budd et al., 2017*). Consistent with the idea that ensheathing glia phagocytose Or42a OSN terminals in a Draper-dependent manner, loss of Draper in ensheathing glia blocks the decrease in the pHluorin/tdTomato ratio observed following critical period EB exposure (*Figure 7E–F*). Together with our previous findings (*Figure 5C–D*), these data

demonstrate that ensheathing glia perform Draper-dependent phagocytosis of Or42a terminals in response to high levels of critical period experience.

## Discussion

Developmental critical periods are defined temporal windows when sensory experience has enduring effects on circuit structure and function. They are remarkable both for the large-scale circuit remodeling seen within them, and for the abruptness with which they open and shut. The *Drosophila* antennal lobe has emerged as an attractive model in which to define cellular and molecular mechanisms underpinning critical period timing and plasticity (*Chodankar et al., 2020*; *Das et al., 2011*; *Devaud et al., 2003a*; *Golovin et al., 2021*; *Golovin et al., 2019*; *Sachse et al., 2007*), as both pre- and post-synaptic sensory neurons can be readily manipulated with robust genetic tools. We focused on the VM7 glomerulus, which contains axonal projections of Or42a olfactory sensory neurons, the OSN type with the greatest sensitivity for ethyl butyrate. Or42a OSN terminals exhibit a striking form of early-life structural plasticity; namely, they shrink dramatically in size following intense early-life EB exposure (*Golovin et al., 2021*; *Golovin et al., 2019*).

Here we elucidate several key features of early-life VM7 plasticity. First, we define an EB concentration (15%v/v in oil) driving elimination of Or42a OSN terminals only when animals are exposed from 0 to 2 DPE. This odor concentration has no effect when presented either immediately before or after. We go on to show that EB exposure from 0 to 2 DPE has long-term consequences for odor processing since both synapse number and synaptic activity remain persistently decreased following cessation of odor exposure. Together, the clearly delineated temporal window and the persistent effect on olfactory circuit function argue that VM7 remodeling is an excellent model in which to investigate underlying mechanisms. We go on to show that Or42a OSN axon terminals are eliminated by ensheathing glia which invade the glomerulus and phagocytose axon terminals via the engulfment receptor Draper. Here we discuss our findings in the broader contexts of Draper-mediated signaling as well as critical period regulation and plasticity.

### Draper drives both ensheathing glial infiltration and synapse engulfment in the critical period

We found that loss of Draper in ensheathing glia completely blocks their ability to invade VM7 and phagocytose Or42a presynaptic terminals in response to critical-period EB exposure. Strikingly, loss of Draper also fully restores spontaneous activity of PNs, arguing that presynaptic terminals remain intact and active until they are engulfed by ensheathing glia. These essential functions of Draper in activity-dependent pruning are similar to the stringent requirement for Draper in the glial injury response (*Doherty et al., 2009*; *MacDonald et al., 2006*; *Ziegenfuss et al., 2012*). In the context of axon injury, Draper is proposed to act upstream of other steps in the engulfment response, including upregulation of phagocytosis genes and extension of glial membranes to injured axons. We propose that Draper-mediated signaling similarly orchestrates multiple components of the glial response during critical period plasticity.

The essential requirement for Draper in experience-dependent pruning during the critical period appears distinct from Draper's role in stereotyped pruning of the mushroom body. During metamorphosis, mushroom body γ lobe axons are engulfed by astrocytes using the Draper and Crk/Mbc/Ced-12 pathways in a partially redundant manner (*Tasdemir-Yilmaz and Freeman, 2014*). Loss of Draper results in a two-day delay in γ lobe pruning, but axons are ultimately pruned (*Awasaki et al., 2006*). Interestingly, astrocyte infiltration has been shown to be a key step in γ lobe axon defasciculation and pruning (*Marmor-Kollet et al., 2023*). It is also noteworthy that we do not observe fragmentation of EB-exposed Or42a OSN terminals in the absence of glial Draper signaling, as evidenced by the normal spontaneous release from Or42a presynaptic terminals in repo-GAL4>UAS- *draper* RNAi flies (*Figure 4F–G*). This contrasts with other pruning models in *Drosophila* such as the larval NMJ and the mushroom body γ, where axonal or synaptic debris accumulates in the absence of Draper (*Fuentes-Medel et al., 2009*; *Hakim et al., 2014*; *Tasdemir-Yilmaz and Freeman, 2014*). For example, γ lobe axon remodeling is proposed to constitute a two-step process of (1) axon fragmentation and (2) clearance of neuronal debris (*Hakim et al., 2014*). Draper's function in debris clearance, but not axon fragmentation, indicates that the γ lobe intrinsic axon fragmentation program is intact in

*draper* null mutants. In contrast, the absence of Or42a physiological changes with loss of glial Draper in critical period EB exposure opens the door to Draper playing a more instructive role in this process. To address this issue, it will be important to assess if Draper overexpression is sufficient to increase the extent of critical period pruning, or whether additional regulatory steps upstream of Draper are involved, as seen previously (*McLaughlin et al., 2019*).

Draper's central role in this critical period plasticity model is consistent with work in mammals demonstrating that the mammalian Draper homolog MEGF10 is crucial for experience-dependent synapse pruning during development and in adults. Astrocytic MEGF10 prunes retinal ganglion cell inputs in the lateral geniculate nucleus and in the adult brain in an activity-dependent manner to refine connectivity across lifespan (*Chung et al., 2013*; *Lee et al., 2021*). Finally, recent work has found that astrocytic MEGF10 is also responsible for thalamo-cortical synapse engulfment in ocular dominance plasticity (*Lee et al., 2022*). Thus, Draper/MEGF10-mediated signaling orchestrates temporally limited critical period pruning in multiple experimental models (*Lee et al., 2022*; *Nelson et al., 2024*).

## A division of labor for glial subtypes in critical period plasticity models?

We found a non-redundant role for Draper-mediated signaling in ensheathing glia. RNAi-mediated *draper* knockdown in ensheathing glia blocks glial infiltration, synapse pruning, and phagocytosis. We and others have shown that Draper is predominantly expressed in the ensheathing glial subtype in the adult antennal lobe (*Doherty et al., 2009*; *Jindal et al., 2023*). The requirement for Draper in ensheathing glia again contrasts with Draper's requirement in mushroom body γ lobe pruning, where it acts in astrocytes (*Hakim et al., 2014*; *Tasdemir-Yilmaz and Freeman, 2014*). We speculate that this reflects a division of labor between astrocytes and ensheathing glia in the adult antennal lobe.

Whereas pupal astrocytes in the antennal lobe highly express Draper, adult astrocytes express it at lower levels (*Doherty et al., 2009*; *Jindal et al., 2023*). While astrocytic processes are highly ramified and intimately associated with antennal lobe synapses, they are not responsible for synaptic pruning in the critical period paradigm discussed here or in the antennal lobe axotomy model. However, while astrocytes do not directly eliminate EB-exposed axon terminals in VM7, it is entirely possible that astrocytes are otherwise involved in critical period plasticity. For example, they may orchestrate the events of phagocytosis by signaling to other glial subtypes, as they do in mammals (*Jha et al., 2019*). It is also possible that antennal lobe astrocytes are involved in setting the temporal bounds of this critical period as they are in other models (*Ackerman et al., 2021*; *Ribot et al., 2021*; *Starkey et al., 2023*). The ease of independently manipulating distinct neuronal and/or glial populations in *Drosophila* using orthogonal binary expression systems (as done in this study) will facilitate the rapid elucidation of relative glial contributions to this form of critical period plasticity.

The well-defined role for a distinct glial subtype described in this study is congruent with recent work from mammalian models. Microglia prune synapses in the contexts of neurodegenerative disease and injury (*Beiter et al., 2024*; *Donat et al., 2017*; *Rajendran and Paolicelli, 2018*), and it is also clear that they employ innate immune signaling pathways to restrain synapse number during development (*Paolicelli et al., 2011*; *Schafer et al., 2012*; *Stevens et al., 2007*; *Wu et al., 2015*). However, perhaps surprisingly, microglia do not employ complement or fractalkine pathways to phagocytose synapses during ocular dominance plasticity (*Schecter et al., 2017*; *Welsh et al., 2020*). Instead, Draper-mediated signaling in astrocytes is responsible for pruning inactive pre- and post-synaptic compartments of thalamo-cortical synapses in the critical period (*Lee et al., 2022*). The differences among these disparate pruning paradigms with respect to glial subtype involved and molecular program engaged demonstrate that synapse pruning is subject to tight, context-specific, regulation.

## Glomeruli exhibit dichotomous responses to critical period odor exposure

By the end of pupariation, the *Drosophila* antennal lobe has attained its characteristic adult morphology and infiltration with astrocyte and ensheathing glial processes, although there is a paucity of data on the degree to which OSNs can detect environmental odorants within the puparium.

Similarly to mammals which are born without sight or hearing, the pupal stage may act as a physical barrier against the opening of experience-dependent plasticity for this and other sensory circuits. For the first week of life, synapses continue to be added such that the antennal lobe is estimated to expand

by roughly 40% over the first 12days of adulthood (*Aimino et al., 2022*; *Devaud et al., 2003a*). The animal's behavioral response to odors matures over the same time frame (*Devaud et al., 2003a*). Perhaps not surprisingly then, a large body of work demonstrates that olfactory circuitry retains plasticity into adulthood (*Chodankar et al., 2020*; *Das et al., 2011*; *Devaud et al., 2003a*; *Devaud et al., 2001*; *Golovin et al., 2019*; *Mallick et al., 2024*; *Sachse et al., 2007*), with the first 48hr of adulthood representing a critical period for several odors, including $CO_2$ and ethyl butyrate. Intriguingly, the cellular targets of plasticity among glomeruli differ. DM2 and DM5 (two EB-responsive glomeruli) as well as V (the lone $CO_2$-responsive glomerulus) increase in size following critical period odor exposure because of elaboration of PN dendrites, likely stemming from strengthening of GABAergic interneuron inputs (*Chodankar et al., 2020*; *Sachse et al., 2007*). In contrast, as described here, another EB- responsive glomerulus, VM7, decreases in size following concentrated odorant exposure resulting from glial phagocytosis of the Or42a OSN axon terminals. While functional consequences have been ascribed to DM2, DM5, and V critical period plasticity (*Chodankar et al., 2020*; *Sachse et al., 2007*), it was unclear whether VM7 remodeling had enduring effects. We now show that while Or42a OSN axon terminal volume recovers, synapse number and postsynaptic PN activity remain decreased for at least 5 days after the end of the exposure, demonstrating long-term consequences of VM7 plasticity during the critical period.

It is unknown whether VM7 remodeling is restricted to Or42a terminals themselves or whether PN and LN terminals are also targets of critical period remodeling in this glomerulus. It has been reported that the overall size of the VM7 PN dendritic arbor is unaltered following chronic EB exposure (*Golovin et al., 2021*). Moving forward, it will be important to test whether the number of VM7 PN and LN synapses change following critical period remodeling using new cell-type-specific labeling techniques for pre- and post-synaptic specializations (*Aimino et al., 2023*; *Parisi et al., 2023*). Regardless of whether VM7 plasticity is strictly confined to its OSNs, VM7 appears distinct among EB-responsive glomeruli in pruning OSN arbors at all (*Figure 2K*). Why might the cellular target(s) of plasticity differ in VM7 and DM2/DM5? It may be a function of the olfactory system's flexibility with regard to EB odor coding. EB is known as a 'public odor' since it binds with varying affinities to multiple odorant receptors (*Münch and Galizia, 2016*). Indeed, EB activates VM7 OSNs at much lower odor concentrations than OSNs in either DM2 or DM5, indicating it has a higher affinity for Or42a (*Kreher et al., 2008*). We propose that Or42a pruning may be a means to efficiently silence these OSNs at high odor concentrations to preserve odor encoding across a wide range of concentrations. Interestingly, Or42b OSNs, which express the closely related Or42b receptor and are also activated at low EB concentrations, are silenced by high EB concentration along much shorter time scales (*Nagel, 2023*; *Tadres et al., 2022*). This is proposed to occur via a reversible 'depolarization block' of Or42b OSNs. We propose that critical period Or42a pruning reflects a more drastic version of the same strategy, irreversibly silencing Or42a inputs when they are unlikely to be needed in the adult's anticipated sensory environment. If so, Or42b OSNs are also predicted to be pruned following chronic exposure to high concentrations of EB. This mechanism may be generalizable to other OSNs expressing high-affinity receptors for public odors, blocking overwhelming input from a single OSN in animals raised in an environment saturated with its cognate odorant. Given that critical periods permit an individual's early experiences to be hard-wired into their neuronal circuitry, activity-dependent pruning may serve to maintain odor encoding in animals raised in a wide array of sensory environments.

## Methods

**Key resources table**

| Reagent type (species) or resource | Designation | Source or reference | Identifiers | Additional information |
|---|---|---|---|---|
| Genetic reagent (*D. melanogaster*) | Or42a-mCD8::GFP | *Golovin et al., 2019* | Broadie lab | |
| Genetic reagent (*D. melanogaster*) | GMR56F03-Gal4 | Bloomington *Drosophila* Stock Center (BDSC) | BDSC Stock #39157; RRID:BDSC_39157; FlyBase ID: FBst0039157 | Genotype: w[1118]; P{GMR56F03-GAL4}attP2 |

Continued

| Reagent type (species) or resource | Designation | Source or reference | Identifiers | Additional information |
|---|---|---|---|---|
| Genetic reagent (D. melanogaster) | UAS-mCD8::RFP | BDSC | BDSC Stock #27398; RRID:BDSC_27398; FlyBase ID: FBst0027398 | Genotype: y¹ w*; P{UAS-mCD8.mRFP.LG}18 a |
| Genetic reagent (D. melanogaster) | UAS-mCD8::GFP | BDSC | BDSC Stock #32186; RRID:BDSC_32186; FlyBase ID: FBst0032186 | Genotype: w*; P{10XUAS-IVS-mCD8::GFP}attP40 |
| Genetic reagent (D. melanogaster) | UAS-luciferase | BDSC | BDSC Stock #35788; RRID:BDSC_35788; FlyBase ID: FBst0035788 | Genotype: y¹ v¹; P{UAS-LUC.VALIUM10}attP2 |
| Genetic reagent (D. melanogaster) | Alrm-Gal4 | *Doherty et al., 2009* | Freeman lab | |
| Genetic reagent (D. melanogaster) | GMR56F03-LexA | BDSC | BDSC Stock #53574; RRID:BDSC_53574; FlyBase ID: FBst0053574 | Genotype: w¹¹¹⁸; P{GMR56F03-lexA}attP40 |
| Genetic reagent (D. melanogaster) | Repo-Gal4 | *Coutinho-Budd et al., 2017* | Coutinho-Budd lab | |
| Genetic reagent (D. melanogaster) | Alrm-LexA | *Coutinho-Budd et al., 2017* | Coutinho-Budd lab | |
| Genetic reagent (D. melanogaster) | LexAop-*draper* RNAi | *Coutinho-Budd et al., 2017* | Coutinho-Budd lab | |
| Genetic reagent (D. melanogaster) | UAS-*draper I* RNAi | *McPhee et al., 2010* | Freeman lab | |
| Genetic reagent (D. melanogaster) | UAS-*draper* RNAi | BDSC | BDSC Stock #36732; RRID:BDSC_36732; FlyBase ID: FBst0036732 | Genotype: y¹ sc* v¹ sev²¹; P{TRiP.HMS01623}attP2 |
| Genetic reagent (D. melanogaster) | Draper::GFP | BDSC | BDSC Stock #63184; RRID:BDSC_63184; FlyBase ID: FBst0063184 | Genotype: y¹ w*; Mi{PT-GFSTF.1}drpr^MI07659-GFSTF.1 |
| Genetic reagent (D. melanogaster) | GMR86E01-Gal4 | BDSC | BDSC Stock #45914; RRID:BDSC_45914; FlyBase ID: FBst0045914 | Genotype: w¹¹¹⁸; P{GMR86E01-GAL4}attP2 |
| Genetic reagent (D. melanogaster) | UAS-pHluorinSE | BDSC | BDSC Stock #82176; RRID:BDSC_82176; FlyBase ID: FBst0082176 | Genotype: w*; P{10XUAS-pHluorin.CAAX}attP2 |
| Antibody | anti-RFP (Rat polyclonal) | ChromoTek | Cat# 5f8; RRID:AB_2336064 | Used 1:600 (IHC) |
| Antibody | anti-GFP (Rabbit polyclonal) | Abcam | Cat# ab6556; RRID:AB_305564 | Used 1:600 (IHC) |
| Antibody | anti-Bruchpilot (Mouse monoclonal) | Developmental Studies Hybridoma Bank | Cat #nc82; RRID:AB_2314866 | Used 1:50 (IHC) |
| Antibody | Goat anti-Mouse IgG (H+L) Highly Cross-Adsorbed Secondary Antibody, Alexa Fluor Plus 647 | Thermo Fisher Scientific | Cat #A32728; RRID:AB_2866490 | Used 1:400 (IHC) |
| Antibody | Goat anti-Rabbit IgG (H+L) Highly Cross-Adsorbed Secondary Antibody, Alexa Fluor 488 | Thermo Fisher Scientific | Cat #A-11034; RRID:AB_2576217 | Used 1:400 (IHC) |
| Antibody | Goat anti-Rat IgG (H+L) Cross-Adsorbed Secondary Antibody, Alexa Fluor 568 | Thermo Fisher Scientific | Cat #A-11077; RRID:AB_2534121 | Used 1:400 (IHC) |

### *Drosophila melanogaster* stocks

Flies were reared on standard cornmeal-molasses-agar medium in a 25 °C incubator on a 12 hr/12 hr light/dark cycle, conditions which were maintained throughout odorant exposure. Unless otherwise noted, *iso31* flies were used as wild-type controls in experimental crosses. Genotypes and reference information for all transgenic fly lines used in this study are provided in the Key Resources Table.

### Odorant exposure

Our odorant exposure protocol was adapted from the method of Golovin and colleagues (*Golovin et al., 2019*; *Nelson et al., 2024*). For critical period exposure (0–2 days post-eclosion), 10 days after beginning crosses in standard bottles of fly food stoppered with cotton plugs, all adults (parents and any progeny that had eclosed) were anesthetized briefly with $CO_2$ and discarded, and the cotton plugs were replaced with a layer of nylon mesh secured with rubber bands. A 1 mL solution of ethyl butyrate (Sigma-Aldrich #E15701) diluted volumetrically in mineral oil (Sigma-Aldrich #M5904; e.g. 150 µL ethyl butyrate was added to 850 µL mineral oil to yield 15% ethyl butyrate), or 1 mL of mineral oil-only control, was prepared in a 1.5 mL microcentrifuge tube, vortexed for 5–10 s, and the open tube was taped to the side of a 3.55 L airtight glass container (Glasslock #MHRB-370). The nylon-covered bottles were placed into the Glasslock containers, which were sealed and returned to the 25 °C incubator. After 24 hr, all adults that eclosed during that time were anesthetized with $CO_2$ and separated by sex and genotype into standard vials of fly food secured with nylon mesh. Vials were returned to the Glasslock containers with fresh odorant solution and placed in the incubator for an additional 48 hr, during which the odorant solution was refreshed after 24 hr. The same procedure was followed for odorant exposures of different timing and duration, with adults collected over a 24 hr period and odorant solutions refreshed every 24 hr.

### Immunohistochemistry

Adult flies were anesthetized with $CO_2$, decapitated, and the heads were fixed for 20 min in ice-cold 4% (v/v) PFA in PBST (1% Triton X-100, 0.01 M phosphate, 0.0027 M KCl, and 0.137 M NaCl), washed three times with PBS, and dissected in ice-cold PBS. After 20 min fixation with 4% PFA in PBST, brains were washed three times with PBS, incubated for 30 min in blocking buffer (5% normal goat serum, 0.3% Triton X-100 in PBS), then incubated for 24 hr at 4 °C with primary antibodies diluted in PBST (0.1% Triton X-100 in PBS). Brains were given three 10 min washes with PBS, then incubated for 24 hr at 4 °C with secondary antibodies diluted in PBST as before. The next day, brains were given three 10 min washes with PBS and kept stationary for 30 min in SlowFade Gold mountant (Thermo Fisher #S36936). Mounting was performed essentially as described (*Kelly et al., 2017*). Unless otherwise noted, all steps were performed with rocking at room temp. A complete list of antibodies and dilutions is provided in the Key Resources Table.

### Confocal microscopy

Images were acquired with a Zeiss LSM 800 confocal laser scanning microscope using a 100 x/1.4 NA Plan-Apochromat objective. Laser intensity and gain were optimized for each z-stack with the exception of fluorescence-based experiments involving pHluorinSE or Mi{MIC}draper::EGFP-FlAsH- StrepII-TEV-3xFlag, in which a single setting was applied to every z-stack.

### Image analysis

Quantification of glomerulus volume and presynaptic content was done using Imaris 9.7.1 (Bitplane). First, the Surfaces function was used on the GFP channel to model the OSN membrane. The following settings were used: baseline subtraction (threshold of 1000) followed by background subtraction (filter width of 10 µm); smoothing enabled (surfaces detail of 0.1 µm); thresholding based on local contrast (largest sphere diameter set to 10 µm), with automatic thresholding enabled; and filter based on number of voxels (cutoff of 500 voxels). The resulting Surfaces were manually processed by a blinded experimenter who used the scissors tool to remove OSN axons prior to the terminal arbor and used the final Surfaces object to mask the Brp channel. The Spots function was then used on the masked channel to quantify the number of Brp puncta within the glomerulus, using the following settings: deconvolution (robust algorithm, standard parameters) followed by background subtraction

(filter width of 10 µm); algorithm set to different Spots sizes (region growing); background subtraction enabled; estimated diameter of 0.4 µm; automated quality filter; Spots regions from absolute intensity, with automatic region thresholding enabled. Quantification of ensheathing glia volume was done in Imaris 9.9.0, with Surfaces created with the LabKit ImageJ plugin (https://imagej.net/plugins/labkit/). A pixel classifier was trained on each image, using the GMR56F03>mCD8::RFP channel by a blinded experimenter; a voxel filter of 50 voxels was used, and no splitting of touching objects was performed. To measure Draper-GFP intensity in the VM7 glomerulus, the Draw function in the Surfaces menu was used to create a shape with 10 vertices and a diameter of 7 µM, approximating a circle. VM7 location in the XY plane and Z-axis bounds were manually determined in each image stack by a blinded experimenter using the Brp channel. Two circles were used to capture VM7: one at the top Z-slice and one at the bottom. The mean fluorescence intensity of the Draper-GFP channel was captured within the volume bounded by the drawn Surface.

## Electrophysiology

Flies removed from odorant exposure for at least two hours were chilled and placed in a dental wax dissecting chamber following isolation of the head. Brains were removed and dissected in a *Drosophila* physiological saline solution (101 mM NaCl, 3 mM KCl, 1 mM $CaCl_2$, 4 mM $MgCl_2$, 1.25 mM $NaH_2PO_4$, 20.7 mM $NaHCO_3$, and 5 mM glucose, with osmolarity adjusted to 235–245 mOsm and pH 7.2), which was pre-bubbled with 95% $O_2$ and 5% $CO_2$. To increase the likelihood of a successful recording, the glial sheath surrounding the brain was focally and carefully removed by using sharp forceps after treating with an enzymatic cocktail of collagenase (0.1 mg/mL), protease XIV (0.2 mg/mL), and dispase (0.3 mg/mL) for 1 min at 22 °C. The surface of the cell body was cleaned with a small stream of saline that was pressure-ejected from a large-diameter pipette under the visualization of a dissecting microscope.

The retrograde loading technique was used to label the postsynaptic PNs of Or42a OSNs. Sharp micropipettes from quartz glass capillaries with a filament (OD/ID: 1.2/0.6 mm) were fabricated with a laser-based micropipette puller (Sutter Instrument #P-2000), backfilled with Calcium Green dextran [5% (v/v) in water] (Thermo Fisher #C6765), and mounted onto an electrode holder. Voltage pulses were generated with a stimulation isolation unit (Model 2200 Analog Stimulus Isolator, A-M Systems) triggered by a TTL pulse with 5ms duration from a USB Multifunction 2 channel Arbitrary Waveform Generator (DigiStim-2, ALA Scientific Instruments). The temporal structural pattern of the TTL pulse was based on Gaussian mixture models (*Tabuchi et al., 2018*). The micropipette was placed in the glomerulus visualized with Or42a-mCD8::GFP fluorescence using a PE300 CoolLED illumination system on a fixed-stage upright microscope (Olympus #BX51WI). The cell bodies of PNs are clustered in the anterodorsal and lateral sides of the antennal lobe (*Jefferis et al., 2001*), so we focused on finding Calcium Green dextran-positive cell bodies within these clusters. LN action potentials have a large amplitude (always >40 mV), which is distinguishable from those of PNs (*Wilson and Laurent, 2005*). Thus, to prevent LNs from being mixed into the dataset, we discarded the recording data having large amplitude spontaneous spikes. We focused on measuring the spontaneous firing activity of PNs, which is driven by peripheral inputs from OSNs (*Gouwens and Wilson, 2009*; *Kazama and Wilson, 2008*).

## Electrophysiological recordings

Perforated patch-clamp recordings were performed essentially as described (*Nguyen et al., 2022*). Patch pipettes (9–12 MΩ) were fashioned from borosilicate glass capillaries (without filament) using a Flaming-Brown puller (Sutter Instrument #P-97) and further polished with a MF200 microforge (WPI) prior to filling with internal pipette solution (102 mM potassium gluconate, 0.085 mM CaCl2, 0.94 mM EGTA, 8.5 mM HEPES, 4 mM Mg-ATP, 0.5 mM Na-GTP, 17 mM NaCl; pH 7.2). Escin was prepared as a 50 mM stock solution in water (stored up to 2 weeks at −20 °C) and was added fresh into the internal pipette solution to a final concentration of 50 µM. Recordings were acquired with an Axopatch 200B (Molecular Devices) and sampled with Digidata 1550B (Molecular Devices) controlled by pCLAMP 11 (Molecular Devices). The voltage signals were sampled at 10 kHz and low-pass filtered at 1 kHz, then analyzed using MATLAB (MathWorks). Spike onset latency was defined as time from threshold to peak depolarization, and the afterhyperpolarization (AHP) decay time constant was determined by the time required for the membrane potential to decrease to 63.2% of its peak value.

## Statistics

Statistical analyses were performed with Prism 10.2.1 (GraphPad). All data is from at least two independent experiments. Sample size was not predetermined. For microscopy experiments, each data point represents one glomerulus; for electrophysiology experiments, each data point represents recordings from one animal. Shapiro-Wilks normality testing showed that our data was not normally distributed, so nonparametric tests were used: the Mann-Whitney test for comparisons of two samples, and the Kruskal-Wallis test with Dunn's test for multiple comparisons for three or more samples. All tests were two-tailed with an alpha of 0.05, and no outliers were excluded. p-values are represented as follows: ns, not significant ($p \geq 0.05$); $*p < 0.05$; $**p < 0.01$; $***p < 0.001$, $****p < 0.0001$.

## Acknowledgements

We are grateful to Kendal Broadie, Marc Freeman, and Mary Logan for fly stocks. We thank the Bloomington Stock Center for fly stocks and the Developmental Studies Hybridoma Bank for antibodies. We thank members of the Broihier lab for helpful discussions and comments on the manuscript. This work was supported by the NIH under the awards R21NS110397 to HTB, R01NS120689 to HTB and JCB; and T32GM152319 to HL, AF, and DJ.

## Additional information

### Funding

| Funder | Grant reference number | Author |
|---|---|---|
| National Institute of Neurological Disorders and Stroke | R01NS120689 | Hans C Leier<br>Alexander J Foden<br>Darren A Jindal<br>Abigail J Wilkov<br>Paola Van der Linden Costello<br>Jaeda Coutinho-Budd<br>Heather T Broihier |
| National Institute of General Medical Sciences | T32GM152319 | Hans C Leier<br>Alexander J Foden<br>Darren A Jindal |
| National Institute of Neurological Disorders and Stroke | R21NS110397 | Hans C Leier<br>Alexander J Foden<br>Darren A Jindal<br>Abigail J Wilkov<br>Paola Van der Linden Costello<br>Heather T Broihier |

The funders had no role in study design, data collection and interpretation, or the decision to submit the work for publication.

### Author contributions

Hans C Leier, Conceptualization, Data curation, Formal analysis, Supervision, Investigation, Visualization, Methodology, Writing – original draft, Writing – review and editing; Alexander J Foden, Conceptualization, Data curation, Supervision, Investigation, Methodology; Darren A Jindal, Investigation, Methodology; Abigail J Wilkov, Paola Van der Linden Costello, Data curation, Formal analysis; Pamela J Vanderzalm, Data curation; Jaeda Coutinho-Budd, Conceptualization, Data curation, Funding acquisition, Investigation, Project administration; Masashi Tabuchi, Formal analysis, Investigation, Writing – original draft; Heather T Broihier, Conceptualization, Formal analysis, Supervision, Funding acquisition, Validation, Visualization, Methodology, Writing – original draft, Project administration, Writing – review and editing

### Author ORCIDs

Hans C Leier https://orcid.org/0000-0002-0363-1444
Alexander J Foden https://orcid.org/0000-0001-7244-619X

Heather T Broihier [ID] https://orcid.org/0000-0003-1363-3088

Reviewer #1 (Public review): https://doi.org/10.7554/eLife.100989.3.sa1
Reviewer #2 (Public review): https://doi.org/10.7554/eLife.100989.3.sa2
Author response https://doi.org/10.7554/eLife.100989.3.sa3

## Additional files

### Supplementary files
MDAR checklist

### Data availability
All data and statistical analyses generated by this study are included in the source data files provided for each figure.

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
