## [Editor Report · eLife Assessment]

Periods in which experience regulates early plasticity in sensory circuits are well established, but the mechanisms that control these critical periods are poorly understood. In this **important** study, the authors examine early-life critical periods that regulate the *Drosophila* antennal lobe and show that constant odor exposure markedly reduces the volume, synapse number, and function of a specific glomerulus. The authors offer **compelling** evidence that these changes are mediated by the invasion of ensheathing glia into the glomerulus where they phagocytose connections via a mechanism involving the engulfment receptor Draper.

---

## [Referee Report · Reviewer #1 (Public review)]

Time periods in which experience regulates early plasticity in sensory circuits are well established, but the mechanisms that control these critical periods are poorly understood. In this manuscript, Leier and Foden and colleagues examine early-life critical periods that regulate the *Drosophila* antennal lobe, a model sensory circuit for understanding synaptic organization. Using early-life (0-2 days old) exposure to distinct odorants, they show that constant odor exposure markedly reduces the volume, synapse number, and function of the VM7 glomerulus. The authors offer evidence that these changes are mediated by invasion of ensheathing glia into the glomerulus where they phagocytose connections via a mechanism involving the engulfment receptor Draper.

This manuscript is a striking example of a study where the questions are interesting, the authors spent a considerable amount of time to clearly think out the best experiments to ask their questions in the most straightforward way, and expressed the results in a careful, cogent, and well-written fashion. It was a genuine delight to read this paper. Overall, this is an incredibly important finding, a careful analysis, and an excellent mechanistic advance in understanding sensory critical period biology.

Comments on latest version:

In the revision, the authors have clearly thought deeply and added provocative new data. They have addressed my concerns and I laud them on an excellent study.

---

## [Referee Report · Reviewer #2 (Public review)]

Sensory experiences during developmental critical periods have long-lasting impacts on neural circuit function and behavior. However, the underlying molecular and cellular mechanisms that drive these enduring changes are not fully understood. In *Drosophila*, the antennal lobe is composed of synapses between olfactory sensory neurons (OSNs) and projection neurons (PNs), arranged into distinct glomeruli. Many of these glomeruli show structural plasticity in response to early-life odor exposure, reflecting the sensitivity of the olfactory circuitry to early sensory experiences.

In their study, the authors explored the role of glia in the development of the antennal lobe in young adult flies, proposing that glial cells might also play a role in experience-dependent plasticity. They identified a critical period during which both structural and functional plasticity of OSN-PN synapses occur within the ethyl butyrate (EB)-responsive VM7 glomerulus. When flies were exposed to EB within the first two days post-eclosion, significant reductions in glomerular volume, presynaptic terminal numbers, and postsynaptic activity were observed. The study further highlights the importance of the highly conserved engulfment receptor Draper in facilitating this critical period plasticity. The authors demonstrated that, in response to EB exposure during this developmental window, ensheathing glia increase Draper expression, infiltrate the VM7 glomerulus, and actively phagocytose OSN presynaptic terminals. This synapse pruning has lasting effects on circuit function, leading to persistent decreases in both OSN-PN synapse numbers and spontaneous PN activity as analyzed by perforated patch-clamp electrophysiology to record spontaneous activity from PNs postsynaptic to Or42a OSNs .

In my view, this is an intriguing and potentially valuable set of data.

Comments on latest version:

After carefully reviewing the revised manuscript, I am satisfied with the authors' responses to my initial suggestions, particularly regarding the synaptic readouts used in their analyses. The authors have clarified their approach with appropriate changes in wording, which enhance the manuscript's clarity and address my previous concerns. Although I believe it could have been beneficial to incorporate postsynaptic markers to further substantiate the findings, I understand this may not have been feasible within the scope of the current study.

Overall, I find that the major claims of the manuscript are now sufficiently supported by the presented data. The revisions have improved the manuscript, and I am confident it meets the standards for publication. I therefore recommend the manuscript for publication in its current form.

---

## [Author Response]

The following is the authors’ response to the original reviews.

**Reviewer #1 (Public Review):**
Time periods in which experience regulates early plasticity in sensory circuits are well established, but the mechanisms that control these critical periods are poorly understood. In this manuscript, Leier and Foden and colleagues examine early-life critical periods that regulate the *Drosophila* antennal lobe, a model sensory circuit for understanding synaptic organization. Using early-life (0-2 days old) exposure to distinct odorants, they show that constant odor exposure markedly reduces the volume, synapse number, and function of the VM7 glomerulus. The authors offer evidence that these changes are mediated by invasion of ensheathing glia into the glomerulus where they phagocytose connections via a mechanism involving the engulfment receptor Draper.This manuscript is a striking example of a study where the questions are interesting, the authors spent a considerable amount of time to clearly think out the best experiments to ask their questions in the most straightforward way, and expressed the results in a careful, cogent, and well-written fashion. It was a genuine delight to read this paper. I have two experimental suggestions that would really round out existing work to better support the existing conclusions and some instances where additional data or tempered language in describing results would better support their conclusions. Overall, though, this is an incredibly important finding, a careful analysis, and an excellent mechanistic advance in understanding sensory critical period biology.

We thank the reviewer for their thoughtful and constructive comments on our manuscript. In response to their critiques, we conducted several new experiments as well as additional analysis and making changes to the text. As requested, we carried out an electrophysiological analysis of VM7 PN firing in *draper* knockdown animals with and without odor exposure. To our surprise, loss of glial Draper fully suppresses the dramatic reduction in spontaneous PN activity observed following critical period ethyl butyrate exposure, arguing that the functional response is restored alongside OSN morphology. It also suggests that the Or42a OSN terminals are intact and functional until they are phagocytosed by ensheathing glia. In other words, glia are not merely clearing axon terminals that have already degenerated. This evidence provides additional support to the claim that the VM7 glomerulus will be an outstanding model for defining mechanism of experience-dependent glial pruning. Detailed responses to the reviewers’ comments follow below.

Regarding the apparent disconnect between the near complete silencing of PNs versus the 50% reduction in Or42a OSN infiltration volume, we agree with the reviewer that this tracks with previous data in the field. While our Imaris pipeline is relatively sensitive, it may not pick up modest changes to terminal arbor architecture. Indeed, as described in Jindal et al. (2023) and in the Methods in this manuscript, we chose conservative software settings that, if anything, would undercount the percent change in infiltration volume. We also note that increased inhibitory LN inputs onto PNs could contribute to dramatic PN silencing we observe. While fascinating, we view LN plasticity beyond the scope of the current manuscript. We removed any mention of ‘silent synapses’ and now speculate about increased inhibition.

**Reviewer #1 (Recommendations For The Authors):**
Major Elements:(1) The authors demonstrate that loss of draper in glia can suppress many of the pruning related phenotypes associated with EB exposure. However, they do not assess electrophysiological output in these experiments, only morphology. It would be great to see recordings from those animals to see if the functional response is also restored.

We performed the experiment the reviewer requested (see Figure 4F-J). We are pleased to report that our recordings from VM7 PNs match our morphology measurements: in repo-GAL4>UAS-*draper* RNAi flies, there was no difference in the innervation of VM7 PNs between animals exposed to mineral oil or 15% EB from 0-2 DPE. This result is in sharp contrast to the near-total loss of OSN-PN innervation in flies with intact glial Draper signaling, and strongly validates the role we propose for Draper in the Or42a OSN critical period.

(2) There is a disconnect between physiology and morphology with a near complete loss of activity from VM7 PNs but a less severe loss of ORN synapses. While not completely incongruent (previous work in the AL showed a complete loss of attractive behavior though synapse number was only reduced 40% - Mosca et al. 2017, eLife), it is curious. Can the authors comment further? Ideally, some of these synapses could be visualized by EM to determine if the remaining synapses are indeed of correct morphology. If not, this could support their assertion of silent inputs from page 7. Further, what happens to the remaining synapses? VM7 PNs should be receiving some activity from other local interneurons as well as neighboring PNs.

We agree that on the surface, our electrophysiology results are more striking than one might expect solely from our measurements of VM7 morphology and presynaptic content. As the reviewer points out, previous studies of fly olfaction have consistently found that relatively modest shifts in glomerular volume in response to prolonged earlylife odorant exposure can be accompanied by drastic changes in physiology and behavior (in addition, we would add Devaud et al., 2003; Devaud et al., 2001; Acebes et al., 2012; and Chodankar et al., 2020, as foundational examples of this phenomenon).

A major driver of these changes appears to be remodeling of antennal lobe inhibitory LNs (see Das et al., 2011; Wilson and Laurent, 2005; Chodankar et al., 2020), especially GABAergic inhibitory interneurons. Perhaps increased LN inhibition of chronically activated PNs, on top of the reduced excitatory inputs resulting from ensheathing glial pruning of the Or42a OSN terminal arbor, would explain the near-total loss of VM7 PN activity we observe after critical period EB exposure. However, given that the scope of our study is limited to critical-period glial biology and does not address the complex topics of LN rewiring or synapse morphology, we have removed the sentence in which we raise the possibility of “silent synapses” in order to avoid confusion. The reviewer is also correct that VM7 PNs have inputs from non-ORN presynaptic partners, including LNs and PNs. So again, perhaps increased inhibitory inputs contributes to the near-complete silencing of the PNs. Given the heterogeneity of LN populations, we view this area as fertile ground for future research.

Language / Data Considerations:(1) Or42a OSNs have other inputs, namely, from LNs. What are they doing here? Are they also affected?

As discussed above, the question of how LN innervation of Or42a OSNs is altered by critical-period EB exposure is an intriguing one that fully deserves its own follow-up study, and we have tried to avoid speculation about the role of LNs when discussing our pruning phenotype. We note at multiple points throughout the text the importance of LNs and refer to previous studies of LN plasticity in response to chronic odorant exposure.

(2) In all of the measurements, what happens to synaptic density? Is it maintained? Does it scale precisely? This would be helpful to know.

We have performed the analysis as requested, which is now included in a supplement to Figure 5. We found that synaptic density shows no trend in variation across conditions and glial driver genotypes.

(3) In Figure 5, the controls for the alrm-GAL4 experiments show a much more drastic phenotype than controls in previous figures? Does this background influence how we can interpret the results? Could the response have instead hit a floor effect and it's just not possible to recover?

The reviewer is correct that following EB exposure, astrocyte vs. ensheathing glial driver backgrounds displayed modest differences in the extent of pruning by volume (0.27 for astros, 0.36 for EG). We note that the two *drpr* RNAi lines that we used had non-significant (but opposite) effects on the estimated size of OSN42a OSN volume in combination with the astrocyte driver, arguing against a floor effect. In addition, a recent publication by Nelson et al. (2024) replicated our findings with a different astrocyte GAL4 driver and *draper* RNAi line. Thus, we are confident that this result is biologically meaningful and not an artifact of genetic background.

(4) The estimation of infiltration measurement in Figure 6 is tricky to interpret. It implies that the projections occupy the same space, which cannot be possible. I'd advocate a tempering of some of this language and consider an intensity measurement in addition to their current volume measurements (or perhaps an "occupied space" measurement) to more accurately assess the level of resolution that can be obtained via these methods.

We completely agree that our language in describing EG infiltration could have been more precise, and we modified our language as suggested. The combination of the Or42a-mCD8::GFP label we and others use, our use of confocal microscopy, and our Surface pipeline in Imaris combine to create a glomerular mask that traces the outline of the OSN terminal arbor, but is nonetheless not 100% “filled” by neuronal membrane and/or glial processes.

(5) Do the authors have the kind of resolution needed to tell whether there is indeed Or42a-positive axon fragmentation (as asserted on p16 and from their data in figures 4, 5, 7). If the authors want to say this, I would advocate for a measurement of fragmentation / total volume to prove it - if not, I would advocate tempering of the current language.

The reviewer brings up a fair criticism: while our assertion about axon fragmentation was based on our visual observations of hundreds of EB-exposed brains, the resolution limits of confocal microscopy do not allow us to rigorously rule out fragmentation within a bundle of OSN axons. Instead, our most compelling evidence for the lack of EB-induced Or42a OSN fragmentation in the absence of glial Draper comes from our new electrophysiology data (Figure 4F-J) in repo-GAL4>UAS-*draper* RNAi animals. We found no difference in spontaneous release from Or42a terminals in flies exposed to mineral oil or 15% EB from 0-2 DPE, which would not be the case if there was Draper-independent fragmentation along the axons or terminal arbors upon EB exposure. We have updated our discussion of fragmentation so that our statements are based on this new evidence, and not confocal microscopy.

(6) There is an interesting Discussion opportunity missed here. Some experiments would, ostensibly, require pupae to detect odorants within the casing via structures consistently in place for olfaction during pupation. It would be useful for the authors to discuss a little more deeply when this critical period may arise and why the experiment where pupae are exposed to EB two days before eclosion and there is no response, occurs as it does. I agree that it's clearly a time when they are not sensitive to the odorant, but that could just be because there's no ability to detect odorants at that time. Is it a question of non-sensitivity to EB or just non-sensitivity to everything?

We share the reviewer’s interest in the plasticity of the olfactory circuit during pupariation, although, as they correctly point out, it is difficult to conceive of an odorant-exposure experiment that could disentangle the barrier effects of puparium from the sensitivity of the circuit itself, and our pre-eclosion data in Figure 3A, D, G does not distinguish between the two. While an investigation into mechanism by which the critical period for ethyl butyrate exposure opens and closes is outside the scope of the present study, we would consider the physical barrier of the puparium to be a satisfactory explanation for why eclosion marks the functional opening of experiencedependent plasticity. As the reviewer suggests, we have added this important nuance to our discussion of the opening of the critical period in the corresponding paragraph of the Results, as well as to the Discussion section “Glomeruli exhibit dichotomous responses to critical period odor exposure.”

Minor Elements:(1) Page 6 bottom: "Or4a-mCD8::GFP" should be "Or42a-mCD8::GFP"(2) Page 15, end of last full paragraph. Remove the "e"

Thank you for pointing out these typos. They have been corrected.

**Reviewer #2 (Public Review):**
Sensory experiences during developmental critical periods have long-lasting impacts on neural circuit function and behavior. However, the underlying molecular and cellular mechanisms that drive these enduring changes are not fully understood. In *Drosophila*, the antennal lobe is composed of synapses between olfactory sensory neurons (OSNs) and projection neurons (PNs), arranged into distinct glomeruli. Many of these glomeruli show structural plasticity in response to early-life odor exposure, reflecting the sensitivity of the olfactory circuitry to early sensory experiences.In their study, the authors explored the role of glia in the development of the antennal lobe in young adult flies, proposing that glial cells might also play a role in experiencedependent plasticity. They identified a critical period during which both structural and functional plasticity of OSN-PN synapses occur within the ethyl butyrate (EB)responsive VM7 glomerulus. When flies were exposed to EB within the first two days post-eclosion, significant reductions in glomerular volume, presynaptic terminal numbers, and postsynaptic activity were observed. The study further highlights the importance of the highly conserved engulfment receptor Draper in facilitating this critical period plasticity. The authors demonstrated that, in response to EB exposure during this developmental window, ensheathing glia increase Draper expression, infiltrate the VM7 glomerulus, and actively phagocytose OSN presynaptic terminals. This synapse pruning has lasting effects on circuit function, leading to persistent decreases in both OSN-PN synapse numbers and spontaneous PN activity as analyzed by perforated patch-clamp electrophysiology to record spontaneous activity from PNs postsynaptic to Or42a OSNs.In my view, this is an intriguing and potentially valuable set of data. However, since I am not an expert in critical periods or habituation, I do not feel entirely qualified to assess the full significance or the novelty of their findings, particularly in relation to existing research.

We thank the reviewer for their insightful critique of our work. In response to their comments, we added additional physiological analysis and tempered our language around possible explanations for the apparent disconnect between the physiological and morphological critical period odor exposure. These changes are explained in more detail in the response to the public review by Reviewer 1 and also in our responses outlined below.

**Reviewer #2 (Recommendations For The Authors):**
I though do have specific comments and questions concerning the presynaptic phenotype they deduce from confocal BRP stainings and electrophysiology.Concerning the number of active zones: this can hardly be deduced from standardresolution confocal images and, maybe more importantly, lacking postsynaptic markers. This particularly also in the light of them speculating about "silent synapses". There are now tools existing concerning labeled, cell type specific expression of acetylcholine-receptor expression and cholinergic postsynaptic density markers (importantly Drep2). Such markers should be entailed in their analysis. They should refer to previous concerning "brp-short" concerning its original invention and prior usage.

We thank the reviewer for their thoughtful approach to our methodology and claims. While the use of confocal microscopy of Bruchpilot puncta to estimate numbers of presynapses is standard practice (see Furusawa et al., 2023; Aimino et al., 2022; Urwyler et al., 2019; Ackerman et al., 2021), the reviewer is correct that a punctum does not an active zone make. Bruchpilot staining and quantification is a well-validated tool for approximating the number of presynaptic active zones, not a substitute for super-resolution microscopy. We made changes to our language about active zones to make this distinction clearer. We have also removed the sentence where we discuss the possibility of “silent synapses,” which both reviewers felt was too speculative for our existing data. Finally, we are highly interested in characterizing the response of PNs and higher-order processing centers to critical-period odorant exposure as a future direction for our research. However, given the complexity of the subject, we chose to limit the scope of this study to the interactions between OSNs and glia.

Regarding their electrophysiological analysis and the plausibility of their findings: I am uncertain whether the moderate reduction in BRP puncta at the relevant OSN::PN synapse can fully account for the significantly reduced spontaneous PN activity they report. This seems particularly doubtful in the absence of any direct evidence for postsynaptically silent synapses. Perhaps this is my own naivety, but I wonder why they did not use antennal nerve stimulation in their experiments?

We refer to previous studies of the AL indicating that moderate changes in glomerular volume and presynaptic content can translate to far more striking alterations in electrophysiology and behavior (Devaud et al., 2003; Devaud et al., 2001; Acebes et al., 2012; and Chodankar et al., 2020, Mosca et al., 2017). This literature has demonstrated that chronic odorant exposure can result in remodeling of inhibitory local interneurons to suppress over-active inputs from OSNs. While we do not address the complex subject of interneuron remodeling in the present study, we find it highly likely that there would be significant changes in interneuron innervation of PNs, independent of glial phagocytosis of OSN excitatory inputs, resulting in additional inhibition. Moving forward, we are very interested in expanding these studies to include odor-evoked changes in PN activity.

Additional minor point: The phrase "Soon after its molecular biology was described (et al., 1999), the *Drosophila melanogaster*" seems somewhat misleading. Isn't the field still actively describing the molecular biology of the fly olfactory system?

We completely agree and have removed this sentence entirely.

Reviewing Editor's Note: to enhance the evidence from mostly compelling in most facets to solid would be to add physiology to the Draper analysis.

These experiments have been completed and are presented in Figure 4F-J.

References

Acebes A, Devaud J-M, Arnés M, Ferrús A. 2012. Central Adaptation to Odorants Depends on PI3K Levels in Local Interneurons of the Antennal Lobe. J Neurosci 32:417–422. doi:10.1523/jneurosci.2921-11.2012

Ackerman SD, Perez-Catalan NA, Freeman MR, Doe CQ. 2021. Astrocytes close a motor circuit critical period. Nature592:414–420. doi:10.1038/s41586-021-03441-2

Aimino MA, DePew AT, Restrepo L, Mosca TJ. 2022. Synaptic Development in Diverse Olfactory Neuron Classes Uses Distinct Temporal and Activity-Related Programs. J Neurosci 43:28–55. doi:10.1523/jneurosci.0884-22.2022

Chodankar A, Sadanandappa MK, VijayRaghavan K, Ramaswami M. 2020. Glomerulus-Selective Regulation of a Critical Period for Interneuron Plasticity in the *Drosophila* Antennal Lobe. J Neurosci 40:5549–5560. doi:10.1523/jneurosci.2192-19.2020

Das S, Sadanandappa MK, Dervan A, Larkin A, Lee JA, Sudhakaran IP, Priya R, Heidari R, Holohan EE, Pimentel A, Gandhi A, Ito K, Sanyal S, Wang JW, Rodrigues V, Ramaswami M. 2011. Plasticity of local GABAergic interneurons drives olfactory habituation. Proc Natl Acad Sci 108:E646–E654. doi:10.1073/pnas.1106411108 Devaud J, Acebes A, Ramaswami M, Ferrús A. 2003. Structural and functional changes in the olfactory pathway of adult *Drosophila* take place at a critical age. J Neurobiol 56:13–23. doi:10.1002/neu.10215

Devaud J-M, Acebes A, Ferrus A. 2001. Odor Exposure Causes Central Adaptation and ´Morphological Changes in Selected Olfactory Glomeruli in *Drosophila*. J Neurosci 21:6274–6282. doi:10.1523/jneurosci.21-16-06274.2001

Furusawa K, Ishii K, Tsuji M, Tokumitsu N, Hasegawa E, Emoto K. 2023. Presynaptic Ube3a E3 ligase promotes synapse elimination through down-regulation of BMP signaling. Science 381:1197–1205. doi:10.1126/science.ade8978

Mosca TJ, Luginbuhl DJ, Wang IE, Luo L. 2017. Presynaptic LRP4 promotes synapse number and function of excitatory CNS neurons. eLife 6:e27347. doi:10.7554/elife.27347

Nelson N, Vita DJ, Broadie K. 2024. Experience-dependent glial pruning of synaptic glomeruli during the critical period. Sci Rep 14:9110. doi:10.1038/s41598-024-59942-3

Urwyler O, Izadifar A, Vandenbogaerde S, Sachse S, Misbaer A, Schmucker D. 2019. Branch-restricted localization of phosphatase Prl-1 specifies axonal synaptogenesis domains. Science 364. doi:10.1126/science.aau9952

Wilson RI, Laurent G. 2005. Role of GABAergic Inhibition in Shaping Odor-Evoked Spatiotemporal Patterns in the *Drosophila* Antennal Lobe. J Neurosci 25:9069–9079.

doi:10.1523/jneurosci.2070-05.2005